# Calcium-driven regulation of voltage-sensing domains in BK channels

**Yenisleidy Lorenzo-Ceballos[1,2], Willy Carrasquel-Ursulaez[2], Karen Castillo[2], Osvaldo Alvarez[2,3], Ramon Latorre[2]\***

[1]Doctorado en Ciencias Mención Neurociencia, Facultad de Ciencias, Universidad de Valparaíso, Valparaíso, Chile; [2]Centro Interdisciplinario de Neurociencia de Valparaíso, Facultad de Ciencias, Universidad de Valparaíso, Valparaíso, Chile; [3]Departamento de Biología, Facultad de Ciencias, Universidad de Chile, Santiago, Chile

**Abstract** Allosteric interactions between the voltage-sensing domain (VSD), the $Ca^{2+}$-binding sites, and the pore domain govern the mammalian $Ca^{2+}$- and voltage-activated $K^+$ (BK) channel opening. However, the functional relevance of the crosstalk between the $Ca^{2+}$- and voltage-sensing mechanisms on BK channel gating is still debated. We examined the energetic interaction between $Ca^{2+}$ binding and VSD activation by investigating the effects of internal $Ca^{2+}$ on BK channel gating currents. Our results indicate that $Ca^{2+}$ sensor occupancy has a strong impact on VSD activation through a coordinated interaction mechanism in which $Ca^{2+}$ binding to a single α-subunit affects all VSDs equally. Moreover, the two distinct high-affinity $Ca^{2+}$-binding sites contained in the C-terminus domains, RCK1 and RCK2, contribute equally to decrease the free energy necessary to activate the VSD. We conclude that voltage-dependent gating and pore opening in BK channels is modulated to a great extent by the interaction between $Ca^{2+}$ sensors and VSDs.
DOI: https://doi.org/10.7554/eLife.44934.001

\*For correspondence:
ramon.latorre@uv.cl

**Competing interests:** The authors declare that no competing interests exist.

## Introduction

Diverse cellular events involve calcium ions as a primary mediator in the signal transduction pathways triggering, among other signaling processes, $Ca^{2+}$-activated conductances. Since the BK channels are regulated by cytosolic $Ca^{2+}$ and depolarizing voltages (*Marty, 1981*; *Pallotta et al., 1981*; *Latorre et al., 1982*), they are integrators of physiological stimuli involving intracellular $Ca^{2+}$ elevation and membrane excitability. BK channels are modular proteins in which each module accomplishes a specific function. Thus, different modules harbor voltage and $Ca^{2+}$ sensors that communicate allosterically with the channel gate (*Cox et al., 1997*; *Horrigan and Aldrich, 1999*; *Horrigan and Aldrich, 2002*; *Horrigan et al., 1999*; *Rothberg and Magleby, 1999*; *Rothberg and Magleby, 2000*; *Cui and Aldrich, 2000*). Functional BK channels are formed by homotetramers of α-subunits (*Shen et al., 1994*), each comprising a transmembrane voltage-sensing domain (VSD) and an intracellular $Ca^{2+}$-sensing C-terminal domain (CTD) that can independently modulate the ion conduction gate in the pore domain (PD) (*Latorre et al., 2017*). The CTDs consist of two non-identical regulators of the conductance of $K^+$ domains (RCK1 and RCK2) arranged in a ring-like tetrameric structure dubbed the gating ring (*Wu et al., 2010*; *Yuan et al., 2010*; *Yuan et al., 2012*; *Hite et al., 2017*; *Tao et al., 2017*). Each RCK domain contains distinct ligand-binding sites capable of detecting $Ca^{2+}$ in the micromolar range (*Schreiber and Salkoff, 1997*; *Bao et al., 2002*; *Xia et al., 2002*).

In the absence of $Ca^{2+}$, the activation of the VSD decreases the free energy necessary to fully open the BK channels in an allosteric fashion (*Horrigan and Aldrich, 1999*; *Horrigan et al., 1999*). With 0 $Ca^{2+}$, very positive membrane potentials are required to drive all voltage sensors to their active conformations (*Cui et al., 1997*; *Stefani et al., 1997*; *Horrigan et al., 1999*; *Contreras et al.,*

*2012*), ultimately leading to near-maximal BK activation. Thus, in cells such as neurons, an appreciable open probability of BK channels at physiologically relevant voltages necessarily requires the activation of the $Ca^{2+}$ sensors on the gating ring. The allosteric interplays established between the functional and structural modules (VSD-PD, CTD-PD, and CTD-VSD) are key for enabling BK channels to operate over a wide dynamic range of internal $Ca^{2+}$ and voltage, thereby fine-tuning the channel's gating machinery. Therefore, understanding the structure-functional bases that underlie the $Ca^{2+}$ and voltage activation mechanisms interrelationship becomes essential to unveil how the channel behaves under different physiological conditions.

The voltage dependence of $Ca^{2+}$-dependent gating ring rearrangements (*Miranda et al., 2013*; *Miranda et al., 2018*) and RCK1 site occupancy (*Sweet and Cox, 2008*; *Savalli et al., 2012*; *Miranda et al., 2018*) as well as the perturbation of VSD movements by $Ca^{2+}$ binding (*Savalli et al., 2012*) support the idea that the energetic interaction between both specialized sensors may be crucial for BK channel activation. The physical CTD-VSD interface has been suggested to provide the structure capable of mediating the crosstalk between these sensory modules and their synergy in activating the pore domain (*Yang et al., 2007*; *Sun et al., 2013*; *Tao et al., 2017*; *Zhang et al., 2017*). However, the strength of the interaction between voltage and $Ca^{2+}$ sensors and their relevance to BK channel activation is still debated (cf. *Horrigan and Aldrich, 2002*; *Carrasquel-Ursulaez et al., 2015*). Also, the functional role that each of the high-affinity $Ca^{2+}$-binding sites plays in the CTD-VSD allosteric interaction is an open question. The RCK1 and RCK2 $Ca^{2+}$-binding sites have distinct functional properties conferred by their different molecular structures and relative positions within the gating ring (*Wu et al., 2010*; *Yuan et al., 2010*; *Yuan et al., 2012*; *Hite et al., 2017*; *Tao et al., 2017*). Thus, the RCK sites differ in their $Ca^{2+}$ binding affinities (*Bao et al., 2002*; *Xia et al., 2002*; *Sweet and Cox, 2008*), divalent cations selectivity (*Oberhauser et al., 1988*; *Schreiber and Salkoff, 1997*; *Zeng et al., 2005*; *Zhou et al., 2012*), voltage dependence (*Sweet and Cox, 2008*; *Savalli et al., 2012*; *Miranda et al., 2018*) and in their contribution to allosteric gating mechanisms (*Yang et al., 2010*; *Yang et al., 2015*). In particular, only the RCK1 site appears to be involved in communicating the $Ca^{2+}$-dependent conformational changes towards the membrane-spanning VSD (*Savalli et al., 2012*; *Miranda et al., 2018*). Recently, the *Aplysia* BK structure has revealed that the N-lobe of the RCK1 domain is in a non-covalent contact with the VSD and the S4-S5 linker that connects the voltage sensor to the pore domain. This RCK1-VSD interaction surface is rearranged when comparing the liganded and $Ca^{2+}$-free structures (*Hite et al., 2017*; *Tao et al., 2017*). In fact, it has been hypothesized that any $Ca^{2+}$-induced rearrangements of the gating ring should ultimately be transmitted to the pore domain via the VSD (*Hite et al., 2017*; *Zhou et al., 2017*). Thereby, defining the extent to which $Ca^{2+}$ binding influences to VSD is central for determining the importance of the crosstalk between sensors in decreasing the free energy necessary to open the BK channel.

Here, we examined the $Ca^{2+}$-dependence of the VSD activation by estimating the allosteric coupling between the $Ca^{2+}$ and voltage sensors. By analyzing gating currents under unliganded and $Ca^{2+}$-saturated conditions, we found a strong energetic influence of $Ca^{2+}$-binding on voltage sensor equilibrium in a manner that is independent of channel opening. These findings show that a major component in the synergistic $Ca^{2+}$ and voltage activation of BK channels resides in the strong coupling between $Ca^{2+}$ binding and the voltage sensor activation. We also found that the $Ca^{2+}$-dependence of voltage sensor activation is consistent with a CTD-VSD allosteric coupling that occurs through a concerted interaction scheme in which each $Ca^{2+}$ bound to one subunit affects all voltage sensors in the BK tetramer equally. Notably, we found that the two distinct RCK1 and RCK2 $Ca^{2+}$ sensors contribute equally to the VSD activation via independent allosteric pathways.

## Results

### Allosteric coupling between $Ca^{2+}$-binding and voltage sensor activation is strong

We characterized the effects of $Ca^{2+}$-binding on voltage sensor activation in BK channels by analyzing the gating currents measured in inside-out patches of *Xenopus laevis* oocyte membrane. The amount of gating charge displaced ($Q_C$) at each $Ca^{2+}$ concentration was obtained by integrating the initial part of the decay of the ON-gating current ($I_G$-ON), which was fitted to a single exponential

(fast ON-gating; see Materials and methods). As we show below, we determined only the gating charge displaced before the opening of the BK channel. *Figure 1A–B* show representative $I_G$-ON records in response to 160 mV voltage step in the nominal absence of $Ca^{2+}$ ('zero' $Ca^{2+}$) and in saturating $Ca^{2+}$ concentration (100 µM $Ca^{2+}$). In *Figure 1A–B*, we also show the initial time courses of the corresponding macroscopic $K^+$ current ($I_K$) activation at the same voltage and internal $Ca^{2+}$ conditions. The $I_G$-ON relaxation exhibits an almost complete decay before the $I_K$ achieves an exponential time course in 0 $Ca^{2+}$ conditions (*Figure 1A*). Thus, the time constant of the $I_K$ activation (~3.4 ms at 160 mV) following a delay of ~160 µs is consistent with the movement of the voltage sensors preceding channel opening. Under saturating internal $Ca^{2+}$ conditions, the $I_G$-ON time course develops 20 times faster than the exponential kinetic of the $I_K$ activation ($\tau_{I_G-ON}$ = 30 µs and $\Delta t_{I_K}$ = 660 µs at 160 mV) and is also almost complete within the time interval comprised by the $I_K$ delay ($\Delta t_{I_K\ (100\ \mu M\ )}$ = 84 µs) (*Figure 1B*). Thus, the fast $I_G$-ON relaxation reflects the movement of the gating charge in the channel's closed conformation, regardless of internal $Ca^{2+}$ concentration.

The $Q_C$ was determined over a wide range of membrane potentials at low and high internal (100 µM) $Ca^{2+}$ concentration. Increasing the $Ca^{2+}$ concentration promoted a large leftward shift ($V_H$ = -142.6 ± 4.5 mV) of the normalized $Q_C$ ($Q_C(V)/Q_{C,\ MAX}$ curves) (*Figure 1C*). In spite of with the large leftward shift of the $Q_C$ at high $Ca^{2+}$ concentrations, we found no appreciable slow component in $I_G$-ON (cf. *Horrigan and Aldrich, 2002*; see also *Figure 2—figure supplement 1A*). As expected, a large $Ca^{2+}$-dependent leftward shift was also observed in the time constants of the exponential decays of $I_G$-ON ($\tau_{I_G-ON}$). The $\tau_{I_G-ON}(V)$ curves were fitted to a two state model ($\tau(V) = 1/(\alpha(V) + \beta(V))$) where the forward ($\alpha$) and backward ($\beta$) rate constants represent the resting-active (R-A) transitions of the voltage sensors, that determine the equilibrium constant of the VSD activation ($J(V) = \alpha(V)/\beta(V)$). The predominant effect of $Ca^{2+}$-binding on VSD activation appears to cause a decrease in the backward rate constant at zero voltage ($\beta_0$; see Materials and methods). $\beta_0$ decreases from 76 ms$^{-1}$ at 0 $Ca^{2+}$ to 7 ms$^{-1}$ at 100 µM $Ca^{2+}$, which results in a shift in the equilibrium of the voltage sensors towards their active conformation. Such a large shift ($\Delta V_H$ = -142.6 ± 4.5 mV) implies that $Ca^{2+}$ binding to the RCK $Ca^{2+}$-binding sites alters the VSD equilibrium, which in consequence causes a decrease in the free energy ($\Delta\Delta G_V^{Ca}$) that defines the voltage sensor R-A equilibrium ($J$) by ~8 kJ/mol ($\Delta\Delta G_V^{Ca}$= -7.98 ± 0.27 kJ/mol). In terms of the allosteric gating scheme (*Horrigan and Aldrich, 2002*), this result means that the $J$ equilibrium constant of VSD becomes amplified by an allosteric factor $E$ equal to 26.4 at $Ca^{2+}$-saturated conditions (*Figure 1D*), revealing a strong allosteric coupling between $Ca^{2+}$-binding sites and voltage sensors.

Families of gating currents ($I_G$) were evoked at different intracellular $Ca^{2+}$ concentrations ([$Ca^{2+}$]$_i$) ranging from 0.1 to 100 µM in $K^+$-free solution (*Figure 2A*). For all experiments, we first measured $I_G$ in 'zero' $Ca^{2+}$ condition and then perfused the internal side with solutions containing different concentrations of $Ca^{2+}$. The increase in internal $Ca^{2+}$ promoted a leftward shift of the $Q_C$ versus voltage ($Q_C(V)$) curves (*Figure 2B-C*), which indicates that $Ca^{2+}$-binding facilitates the activation of the voltage sensor, being more prominent as $Ca^{2+}$-binding site occupancy increases.

*Figure 2A* shows that the OFF-gating current was dramatically modified becoming smaller in amplitude and with slower kinetics as the internal $Ca^{2+}$ concentration was increased (see also *Figure 2—figure supplement 1A* and *Figure 2—figure supplement 2A*). At least two components (fast and slow components) could be resolved in the OFF gating current decay in 'zero' and 10 µM internal $Ca^{2+}$ conditions (*Figure 2—figure supplement 1B-C*). In *Figure 2—figure supplement 1D*, we fitted $\tau_{I_G-ON}(V)$ data for 0 and 10 µM internal $Ca^{2+}$ to a two-state R-A model. *Figure 2—figure supplement 1D* shows that the time constants of the fast relaxation of the $I_G$-OFF evoked at -90 mV (open and closed orange circles) are in reasonable agreement with the time constants values extrapolated to the same voltage from the two-state model of the $I_G$-ON kinetic. The change in the time constant of the fast component of OFF-gating current detected at 10 µM $Ca^{2+}$ relative to 'zero' $Ca^{2+}$ condition agrees with a $Ca^{2+}$-induced effect primarily on the backward rate of the R-A transition ($\beta_0$ decreases from 76 ms$^{-1}$ at 0 $Ca^{2+}$ to 10.6 ms$^{-1}$ at 10 µM $Ca^{2+}$, see *Figure 2—figure supplement 1D*), which is consistent with the large $Q_C(V)$ shift observed ($\Delta V_{H\ (10\ \mu M)}$ = -107.1 ± 17.1 mV). On the other hand, the relative contribution of the slower component increasing as internal $Ca^{2+}$ is increased, reflecting an increase in the open probability of the channel (*Figure 2—figure supplement 1E-F*). This kinetic behavior recapitulates the effect described on gating charge displacement as a function of the depolarizing pulse duration (*Horrigan and Aldrich, 2002*;

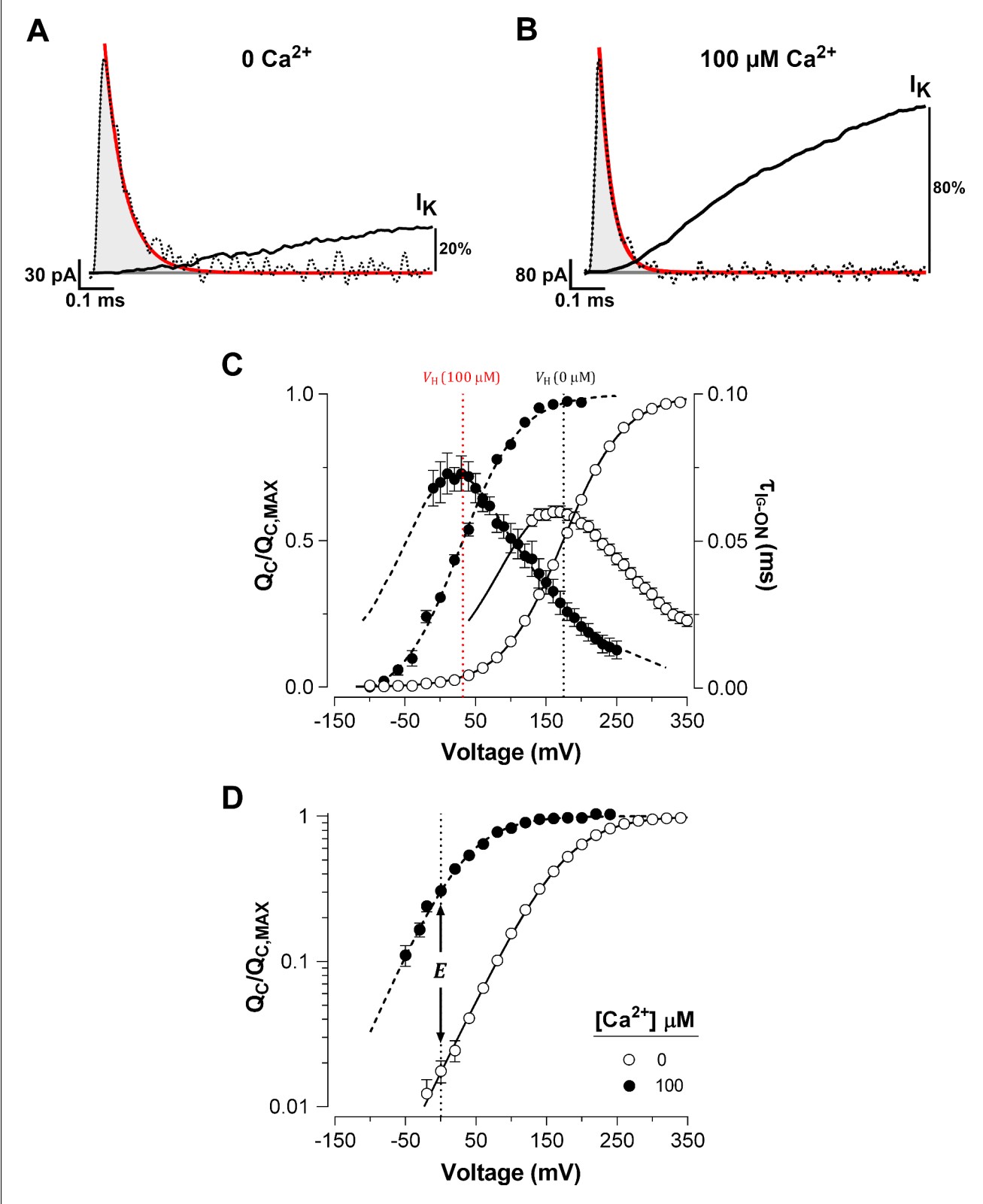

**Figure 1.** Ca$^{2+}$ binding strongly affects the activation of VSD in BK channels. (A–B) Fast component of the ON-gating current ($I_G$-ON) at 0 and 100 μM internal Ca$^{2+}$ concentration, respectively. The representative $I_G$-ON records were evoked by applying a 160 mV voltage step of 1 ms duration. The first 100 μs of the $I_G$-ON were fitted to a single exponential function (red line; $\tau_{I_G-ON\ (0\ Ca^{2+})}$ = 58 μs and $\tau_{I_G-ON\ (100\ \mu M)}$ = 30 μs). The area under the curve described by the monoexponential fit (gray areas) was integrated to obtain the charge displaced between closed states ($Q_C$). For comparison, the initial

*Figure 1 continued on next page*

*Figure 1 continued*

time course of the macroscopic K$^+$ current ($I_K$) activation (solid black line) obtained at 0 and 100 μM Ca$^{2+}$ were superimposed on the $I_G$-ON under the same internal Ca$^{2+}$ conditions. The $I_K$ were evoked by a 15 ms pulse to 160 mV. The $I_K$ is described by an exponential function with initial delay: $\Delta t_{I_K\ (0\ Ca^{2+})} = 0.16$ ms and $\tau_{I_K\ (0\ Ca^{2+})} = 3.37$ ms at 0 Ca$^{2+}$; and $\Delta t_{I_K\ (100\ \mu M)} = 0.08$ ms and $\tau_{I_K\ (100\ \mu M)} = 0.66$ ms at 100 μM Ca$^{2+}$. After 1 ms, the $I_K$ increased to about 20% and 80% of its steady-state amplitude in 0 Ca$^{2+}$ and 100 μM Ca$^{2+}$, respectively. (C) Voltage-dependence of the $Q_C$ and of the gating current time constants ($\tau_{I_G-ON}$) at 0 Ca$^{2+}$ (open circles) and 100 μM Ca$^{2+}$ (filled circles). Gating charge-voltage relationships ($Q_C(V)/Q_{C,\ MAX}$) were obtained by integrating the fast component for each ON $I_G$ trace (from −90 mV to 350 mV). Boltzmann fitting to the experimental data (mean ± SEM) is indicated by solid line at 'zero' Ca$^{2+}$ ($V_H = 174.5 \pm 2.4$ mV, $z_Q = 0.60 \pm 0.01$, $n = 25$) and by a dashed line at 100 μM Ca$^{2+}$ ($V_H = 31.9 \pm 4.5$ mV, $z_Q = 0.66 \pm 0.01$, $n = 7$). Right ordinate shows the time constants data (mean ± SEM) of the exponential decays of $I_G$-ON ($\tau_{I_G-ON}$) plotted against the voltage. The best fit to a two-state model of the VSD activation where the $z_J$ was constrained to the values found for the $Q_C(V)$ relation ($z_Q$) is indicated by solid lines at 'zero' Ca$^{2+}$ ($\alpha_0 = 3.73$ ms$^{-1}$, $\beta_0 = 76.10$ ms$^{-1}$ and $\delta = 0.29$) and dashed line at 100 μM Ca$^{2+}$ ($\alpha_0 = 7.28$ ms$^{-1}$, $\beta_0 = 6.98$ ms$^{-1}$ and $\delta = 0.36$). The corresponding $V_H$ at each internal Ca$^{2+}$ concentration is indicated by a vertical line. (D) Semi-logarithmic plot of the $Q_C(V)/Q_{C,\ MAX}$ curves at 0 Ca$^{2+}$ and saturating Ca$^{2+}$ concentration (100 μM). The allosteric parameter $E$ determines the vertical displacement of the 0 mV intercept (dashed vertical line) of the $Q_C(V)/Q_{C,\ MAX}$ curve at 100 μM Ca$^{2+}$ relative to the 0 Ca$^{2+}$ condition.

DOI: https://doi.org/10.7554/eLife.44934.002

*Carrasquel-Ursulaez et al., 2015*; *Contreras et al., 2012*), and confirms that this phenomenon is associated with the time course of channel opening revealing the allosteric interaction between voltage sensors and the pore gate (*Horrigan and Aldrich, 2002*).

Since the OFF gating charge cannot be accurately estimated from the OFF gating current recorded at -90 mV, we performed two experiments using 100 μM internal Ca$^{2+}$ concentration and recorded the OFF gating currents at -150 mV and -200 mV (*Figure 2—figure supplement 2A*). At these applied voltages, the charge displaced in the ON is recovered in the OFF (*Figure 2—figure supplement 2B*). As expected from the large shift to the left along the voltage axis of $Q_C$ at saturating internal Ca$^{2+}$ concentrations, there was no difference between the $Q_C$ curve and the gating charge-voltage curves obtained from the OFF gating currents 2 ms after the onset of the voltage pulse, for applied voltages of -150 or -200 mV (*Figure 2—figure supplement 2C*).

## Ca$^{2+}$ binding to a single α-subunit affects the R-A voltage sensor equilibrium of all four subunits equally

Taking advantage of the dose-dependent effect of Ca$^{2+}$ on voltage sensor activation we investigated the underlying mechanism of the communication between the Ca$^{2+}$ binding and voltage sensors in the context of the well-established Horrigan-Aldrich (HA) allosteric gating model (*Horrigan and Aldrich, 2002*). Two different mechanisms were proposed by Horrigan and Aldrich for the interaction between the Ca$^{2+}$-binding sites and voltage sensors. The first mechanism proposes that Ca$^{2+}$ binding to one α-subunit only affects the VSD in the same subunit (Scheme I) (*Figure 3A*), whereas the second mechanism proposes that the Ca$^{2+}$ binding affects all four VSD equally (Scheme II) (*Figure 3B*). It should be noted that the standard HA model makes two simplifying assumptions. First, the model considers that there is a single Ca$^{2+}$-binding site per α-subunit. Second, the model assumes the Scheme I as the Ca$^{2+}$ binding-VSD interaction scheme underlying the general gating mechanism of BK channel (*Horrigan and Aldrich, 2002*).

We simulated the normalized $Q_C(V)$ curves over a wide range of Ca$^{2+}$ concentrations (from 0 to 10 mM) for each Ca$^{2+}$-VSD interaction scheme (*Figure 3C-D*). We assumed that the fast gating currents measured correspond to the charge displaced by the R-A transitions and does not include the charge associated with the transition between the activated states (see *Figure 3—figure supplement 1B-C*). In other words, *Equations 4 and 6* given in the Appendix describing the $Q_C(V)$ relationships for Schemes I and II are based on the assumption that channels equilibrate between Ca$^{2+}$-bound states while the channel is closed and voltage sensors are not activated. We further assume that this distribution is not altered when gating current is measured because Ca$^{2+}$ binding is slower than voltage-sensor activation. These assumptions are reasonable since the Ca$^{2+}$-binding rate constant estimated for BK is $1.8 \times 10^8$ M$^{-1}$s$^{-1}$ (*Hou et al., 2016*) implying that at 10 μM internal Ca$^{2+}$ (the highest non-saturating Ca$^{2+}$ concentration tested) the time constant of the Ca$^{2+}$ binding is ~300 μs, while the VSD activates with a time constant of ~60 μs at $V = V_H$ (see the Appendix for details of the simulations).

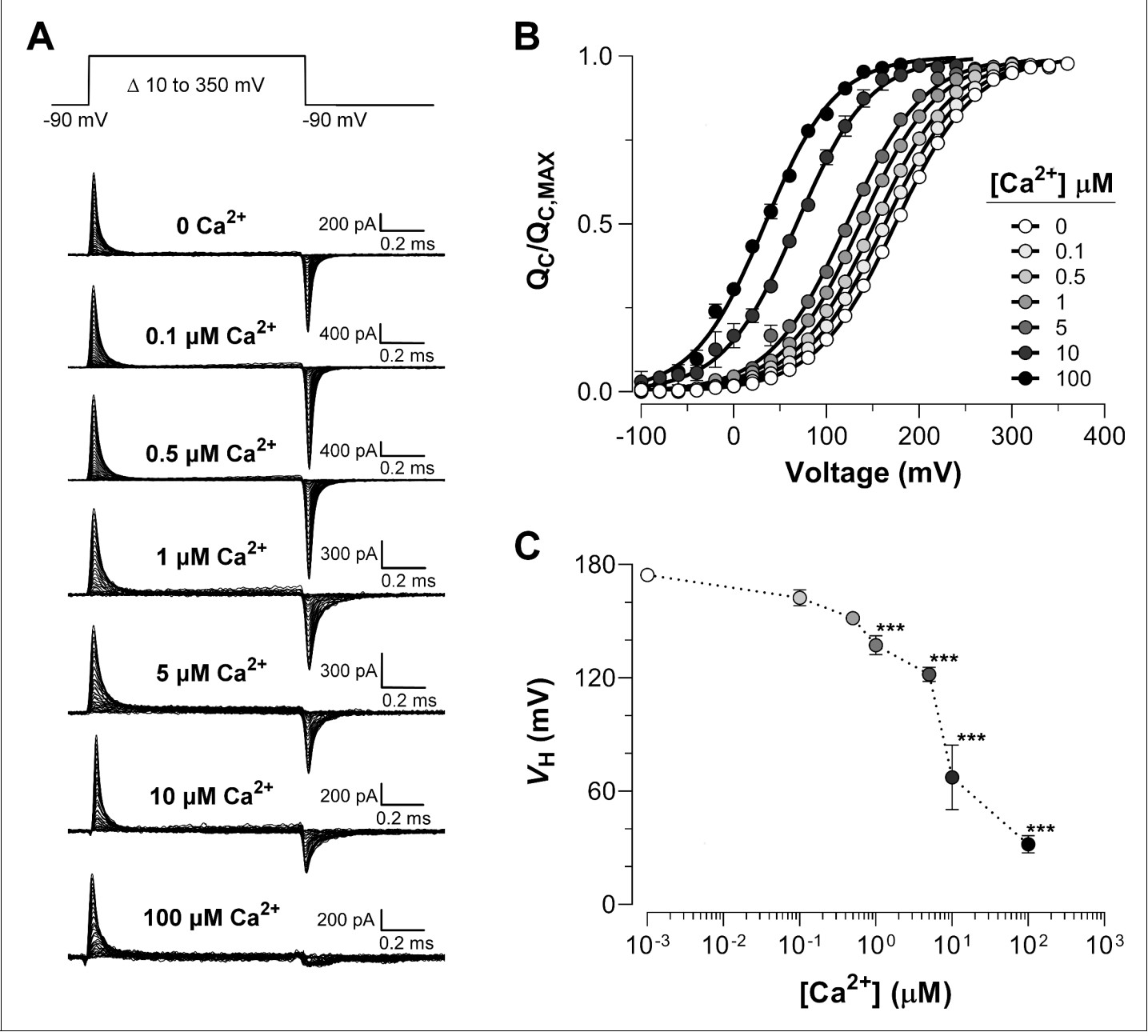

**Figure 2.** $Ca^{2+}$-dependent effects on VSD activation in BK channels. (**A**) Representative gating current ($I_G$) recordings at different internal $Ca^{2+}$ concentrations (from 0 to 100 μM). $I_G$ was evoked by the indicated voltage protocol of 1 ms duration. Representative gating current records are from different patches with the exception of 0 and 100 μM $Ca^{2+}$. (**B**) Gating charge-voltage relationships ($Q_C(V)$) were obtained by integrating the fast component for each ON $I_G$ trace. Normalized gating charge data ($Q_C(V)/Q_{C, MAX}$) (mean ± SEM) were fitted using a single Boltzmann function (solid lines). The fit parameters are as follows: 'zero' $Ca^{2+}$ ($V_H$ = 174.5 ± 2.4 mV, $z_Q$ = 0.60 ± 0.01, $n$ = 25); 0.1 μM $Ca^{2+}$ ($V_H$ = 162.4 ± 4.2 mV, $z_Q$ = 0.59 ± 0.01, $n$ = 5); 0.5 μM $Ca^{2+}$ ($V_H$ = 151.6 ± 1.3 mV, $z_Q$ = 0.60 ± 0.01, $n$ = 5); 1 μM $Ca^{2+}$ ($V_H$ = 137.1 ± 5.1 mV, $z_Q$ = 0.61 ± 0.01, $n$ = 5); 5 μM $Ca^{2+}$ ($V_H$ = 121.9 ± 3.8 mV, $z_Q$ = 0.63 ± 0.01, $n$ = 5); 10 μM $Ca^{2+}$ ($V_H$ = 67.3 ± 17.1 mV, $z_Q$ = 0.64 ± 0.08, $n$ = 4); 100 μM $Ca^{2+}$ ($V_H$ = 31.9 ± 4.5 mV, $z_Q$ = 0.66 ± 0.01, $n$ = 7). (**C**) $V_H$ obtained from the $Q_C(V)$ curves as a function of $Ca^{2+}$ concentration (mean ± SEM). $Ca^{2+}$ binding produces a leftward shift in $V_H$ ($\Delta V_H$): 0.1 μM $Ca^{2+}$ ($\Delta V_H$ = −12.1 ± 3.5 mV); 0.5 μM $Ca^{2+}$ ($\Delta V_H$ = −22.9 ± 1.8 mV); 1 μM $Ca^{2+}$ ($\Delta V_H$ = −37.1 ± 3.5 mV); 5 μM $Ca^{2+}$ ($\Delta V_H$ = −50.3 ± 4.7 mV); 10 μM $Ca^{2+}$ ($\Delta V_H$ = −107.1 ± 17.1 mV); 100 μM $Ca^{2+}$ ($\Delta V_H$ = −142.6 ± 4.5 mV). One-way ANOVA followed by Dunnett's post-hoc test analysis was used to assess statistical significance of the $Ca^{2+}$-induced shifts in $V_H$ (***$p<0.001$).

DOI: https://doi.org/10.7554/eLife.44934.003

The following figure supplements are available for figure 2:

**Figure supplement 1.** $Ca^{2+}$ increases the slow component of the OFF gating currents.

*Figure 2 continued on next page*

*Figure 2 continued*

DOI: https://doi.org/10.7554/eLife.44934.004

**Figure supplement 2.** Voltage-dependent movement of the ON and OFF gating charge has a similar behavior at saturating Ca²⁺ conditions.
DOI: https://doi.org/10.7554/eLife.44934.005

At extreme conditions of low (0.03 to 0.1 µM) and high ($\geq$100 µM) internal Ca²⁺, VSD activation behaves in a mechanism-independent manner since all voltage sensors are in the same functional state (unliganded or saturated). However, the distinctive effects on $Q_C(V)$ curves at intermediate Ca²⁺ concentrations (1-10 µM) provide useful signatures to distinguish between the two mechanisms. Indeed, Scheme I predicts two functional states of the VSD depending on the occupancy status of the Ca²⁺ site (Ca²⁺ bound and unbound) such that the $Q_C(V)$ curve is described by the fractional distribution of the unliganded and Ca²⁺-saturated functional states like an all-or-none allosteric effect (*Figure 3C*; *Figure 3—figure supplement 1B* and *Equation 4* in Appendix). By contrast, the Ca²⁺-binding effect on the VSD activation according to Scheme II is characterized by a five-component Boltzmann function (*Figure 3—figure supplement 1C* and *Equation 6* in Appendix). Each component represents a single functional state determined by the number of Ca²⁺ bound to the channel (from 0 to 4). In such a case, the $Q_C(V)$ curves resulting from a distribution of functional states are equivalent to a single Boltzmann function, leftward shifted by an incremental allosteric effect (from $E$ to $E^4$) as the number of Ca²⁺ bound to the channel increases (*Figure 3D*). The experimental leftward shift of the $Q_C(V)$ curves occur with constant slope ($z_Q$) (*Figure 2B*), which is consistent with Model II. It can be argued that Scheme I can produce single Boltzmann curves without a change in $z_Q$ if Ca²⁺ re-equilibration is fast enough. However, *Figure 3—figure supplement 2* demonstrates that to recover the experimental $z_Q$ requires a Ca²⁺ binding rate constant that is 100-fold faster than the one reported by *Hou et al. (2016)* exceeding the diffusion limit constraint. Therefore, we can conclude that Ca²⁺-binding is slow enough to safely ignore calcium re-equilibration during the 100 µs it takes to measure the gating charges.

To further elucidate the mechanism by which Ca²⁺ and voltage sensors interact, we performed fits of the $Q_C(V)$ data using the two different models represented by Scheme I and Scheme II (*Figure 4A-B*). The allosteric factor $E$ that accounts for the coupling between the Ca²⁺-binding sites and the voltage sensors was constrained to values calculated from the experimental data for the $Q_C(V)$ shift at Ca²⁺ saturating conditions (100 µM) in relation to the same curve in the absence of Ca²⁺. The $z_J$, $J_0$ and $K_D$ parameters obtained during the fitting procedure were very similar for each model (*Table 1*). The fitted values for the affinity constant ($K_D$ = 6 µM) agree with previous reports (*Horrigan and Aldrich, 2002*; *Cox et al., 1997*) although they are slightly smaller than those estimated for the closed conformation of the channel ($K_D$ = 11 µM). Based on the Akaike Information Criterion (AIC) (*Akaike, 1974*), the model that fits to the $Q_C(V, [\text{Ca}^{2+}])$ data best is Model II, whereas the probability of the Model I being the best model is negligible ($w_i = 10^{-47}$) (*Table 1*). Both models generates a $V_H$-log[Ca²⁺] curve that accounts reasonably well for the dose-response experimental data (*Figure 4C*). However, Model I predicts a pronounced decrease in the $z_Q$ parameter of the $Q_C(V)$ curves at intermediate Ca²⁺ conditions which differs markedly from the experimental values (*Figure 4D*). This prediction is a consequence of the fractional distribution of two very distinctive functional states of the voltage sensors (unliganded and Ca²⁺-saturated, see above). Additionally, the behavior of $Q_C(V)$ curves at intermediate Ca²⁺ concentrations (1-10 µM) is qualitatively consistent with the phenotype exhibited by the Ca²⁺-VSD scheme II (*Figure 3D* and *Figure 4B*). Thus, the experimental dose-dependent effect of Ca²⁺ on voltage sensor activation suggests that Ca²⁺-binding to a single α-subunit of BK channels increases $E$-fold the equilibrium constant $J$ that defines the equilibrium between resting and active conformations of the voltage sensors in all four subunits.

## High-affinity Ca²⁺-binding sites in RCK1 and RCK2 domains contribute equally to the allosteric coupling between Ca²⁺ and voltage sensors

Under physiological conditions, the RCK1 and RCK2 high-affinity Ca²⁺-binding sites are responsible for all the calcium sensitivity of the activation of BK channel (*Schreiber and Salkoff, 1997*; *Bao et al., 2002*; *Bao et al., 2004*; *Xia et al., 2002*). However, distinct physiological roles of the RCK1 Ca²⁺-sensor and Ca²⁺ bowl may be based on their functionally and structurally distinctive

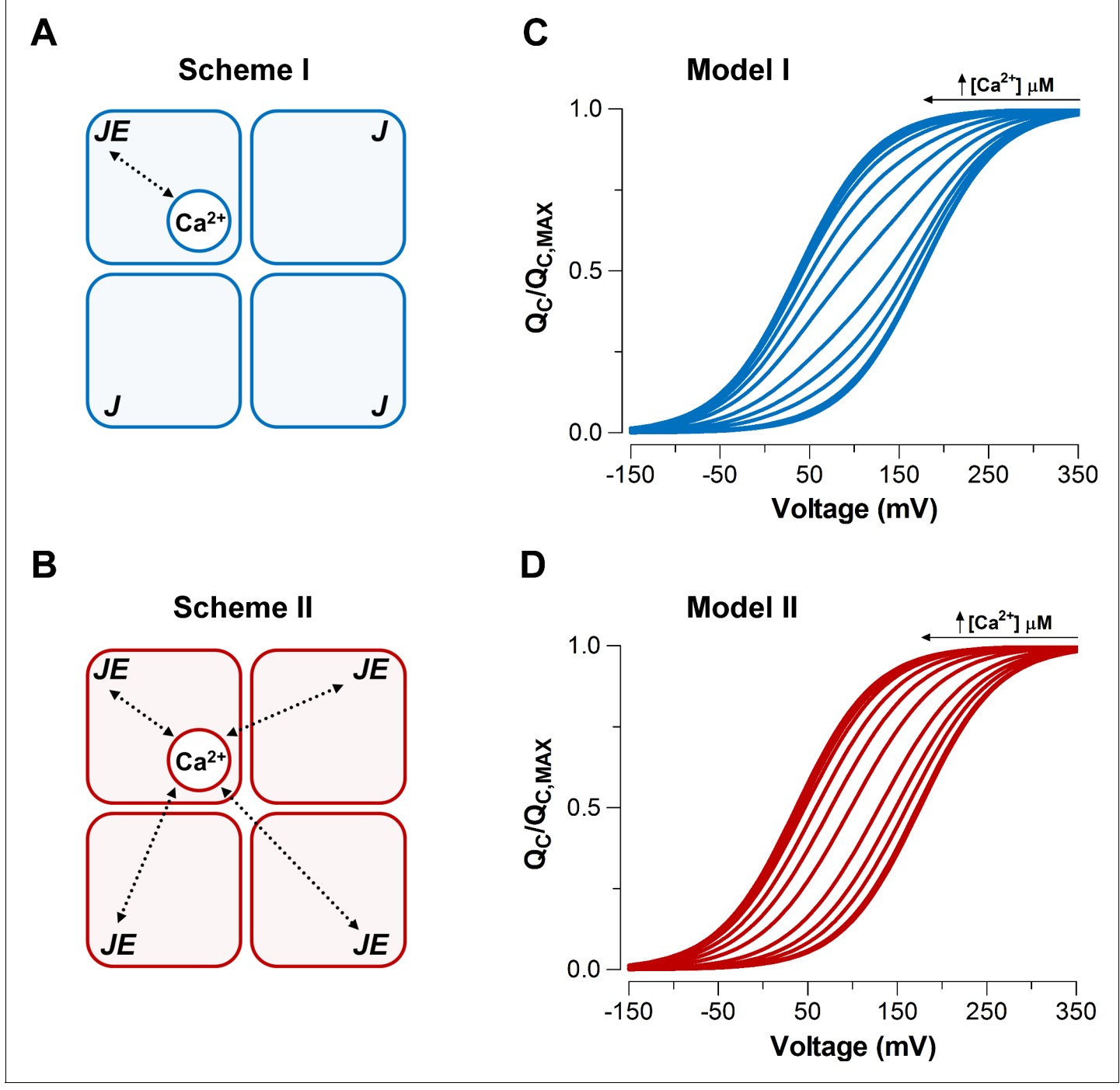

**Figure 3.** Model-dependent behavior of the $Q_C(V)$ curves based on the CTD-VSD interaction mechanisms according to the fractional occupancy of Ca$^{2+}$-binding sites. (A–B) Cartoons representing two interaction schemes between voltage sensors and Ca$^{2+}$-binding sites (modified from *Horrigan and Aldrich, 2002*). Scheme I (A) assumes that Ca$^{2+}$-binding only affects the voltage sensor of one α-subunit ($E_{M1}$), whereas Scheme II (B) predicts that binding of Ca$^{2+}$ to one α-subunit will affect VSD in all subunits equally, increasing the voltage sensor equilibrium constants ($J$) $E_{M2}$-fold in all four subunits ($E_{M2}^4$, when the four Ca$^{2+}$ sites are occupied). In both schemes, a single Ca$^{2+}$-binding site is considered in each α-subunit. (C–D) Predictions of $Q_C(V)$ relationships at different internal calcium concentration (from 0 to 10 mM) by two distinctive interaction mechanisms between Ca$^{2+}$-binding sites and voltage sensors (Scheme I and Scheme II), respectively. $Q_C(V)$ curves were generated using *Equation 4* (blue: Model I) or *Equation 6* (red: Model II), and the following set of parameters: $z_J$ = 0.58, $J_0$ = 0.018, $K_D$ = 11 μM and $E_{M1}$ = $E_{M2}^4$ = 25.

DOI: https://doi.org/10.7554/eLife.44934.006

The following figure supplements are available for figure 3:

*Figure 3 continued on next page*

*Figure 3 continued*

**Figure supplement 1.** Kinetic models of the VSD activation according to the CTD-VSD interaction schemes.
DOI: https://doi.org/10.7554/eLife.44934.007
**Figure supplement 2.** The resting-active transition of the voltage sensor is fast enough to assume no Ca$^{2+}$ re-equilibration during gating currents measurements.
DOI: https://doi.org/10.7554/eLife.44934.008

properties (*Zeng et al., 2005*; *Sweet and Cox, 2008*; *Yang et al., 2010*; *Savalli et al., 2012*; *Tao et al., 2017*). We next asked what the energetic contribution to VSD equilibrium was of the two high-affinity Ca$^{2+}$-binding sites contained in the RCK1 and RCK2 domains.

To elucidate the effect of each Ca$^{2+}$-sensor on the VSD activation we used mutations that selectively and separately abolish the function of the two different RCK Ca$^{2+}$-sites. Disruption of the RCK1 Ca$^{2+}$-sensor by the double mutant D362A/D367A (*Xia et al., 2002*) significantly reduced (48%, $\Delta V_{H(D362A/D367A)}$ = -74.9 ± 4.7 mV) the leftward shift of the $Q_C(V)$ curves at 100 μM Ca$^{2+}$ compared with that for the wild-type (WT) BK channel (*Figure 5A, C*). We also examined the effect of the mutant M513I (*Bao et al., 2002*) which has been shown to eliminate the Ca$^{2+}$ sensitivity mediated by the RCK1 site (*Bao et al., 2002*; *Bao et al., 2004*; *Zhang et al., 2010*). In this mutant, the 100 μM Ca$^{2+}$-induced shift in $V_H$ of the VSD activation curve was also considerably smaller compared to that for the WT channel (about 54%, $\Delta V_{H(M513I)}$ = -65.4 ± 2.6 mV) (*Figure 6*). Therefore, both mutations affect the Ca$^{2+}$-induced enhancement of the activation of the voltage sensor very similarly through the RCK1 site (*Figure 6C*), although their mechanisms of action could be quite different. The M513 residue appears to participate in the stabilization of the proper conformation of the RCK1 Ca$^{2+}$-site whereas D367 is a key residue in the coordination of Ca$^{2+}$ ions (*Wu et al., 2010*; *Zhang et al., 2010*; *Tao et al., 2017*). On the other hand, neutralization of the residues forming part of the Ca$^{2+}$ bowl (*Schreiber and Salkoff, 1997*) (5D5A mutant, see Materials and methods) on the RCK2 domain decreased the leftward shift of the $Q_C(V)$ curve by approximately 54% ($\Delta V_{H(5D5A)}$ = -65.7 ± 4.7 mV) when Ca$^{2+}$ was increased up to 100 μM (*Figure 5B, D*). The effect of Ca$^{2+}$ binding on $V_H$ contributed by each high-affinity Ca$^{2+}$ site is roughly half that for WT channels with both sites intact (*Figure 5E*). Therefore, both high-affinity Ca$^{2+}$-binding sites contribute approximately equally to the decrease in free energy that is necessary to activate the VSD. Indeed, the change of free energy of the resting-active equilibrium of the voltage sensor in response to Ca$^{2+}$-binding at RCK2 site is ~ -4 kJ/mol ($\Delta\Delta G_V^{Ca}$(D362A/D367A) = -4.2 ± 0.3 kJ/mol and $\Delta\Delta G_V^{Ca}$(M513I) = -3.6 ± 0.5 kJ/mol) (*Figure 5C* and *Figure 6C*). Similarly, the occupation of the RCK1 Ca$^{2+}$-binding site decreased the free energy necessary to activate the VSD in -3.8 ± 0.4 kJ/mol ($\Delta\Delta G_V^{Ca}$(5D5A)). Remarkably, these findings reveal an additive effect of Ca$^{2+}$-binding to the RCK1 and Ca$^{2+}$ bowl sites on the VSD activation, which suggest independent allosteric pathways through which they exert their modulation on the VSD.

Taking these results into account, we expanded the Ca$^{2+}$-VSD interaction models considering the energetic contribution of the two kinds of Ca$^{2+}$ sensors on the VSD per α-subunit ($E_{WT} = E_{S1} * E_{S2}$) (*Figure 3—figure supplement 1D*). As described in the above model fittings, the allosteric factors $E$ for each one RCK1 and RCK2 sites ($E_{S1}$ and $E_{S2}$) were constrained to values equivalent to the Ca$^{2+}$-induced energetic perturbations of the voltage sensor equilibrium for the 5D5A and D362A/D367A mutants, respectively. The inclusion of two Ca$^{2+}$ sites per subunit in the Ca$^{2+}$-VSD interaction schemes increases the functional states of the VSD for both models (*Figure 3—figure supplement 1E*). Therefore, it becomes difficult to discriminate between different models based on the phenotype of the $Q_C(V)$ curves at intermediate Ca$^{2+}$ concentrations (*Figure 4—figure supplement 1A-B*). However, according to the AIC criteria (compare AICs for the models in *Table 1*), Model II - Two Ca$^{2+}$ sites is the model that best describes the $Q_C(V, [Ca^{2+}])$ data ($w_i$= 0.99). As mentioned when describing the one Ca$^{2+}$ site models, the goodness of the different models can be better appreciated by the Ca$^{2+}$-dependence of the $z_Q$ parameter (*Figure 4—figure supplement 1C-D*). The fractional distribution of the distinct functional states of the VSDs defined by Model I -Two Ca$^{2+}$ sites (unliganded, RCK1 site occupied, RCK2 site occupied, and the two sites occupied; see *Figure 3—figure supplement 1E*) produce a $z_Q$-log[Ca$^{2+}$] curve with a pronounced minimum when the Ca$^{2+}$ concentration is close to the $K_D$ value (*Figure 4—figure supplement 1C*). Thus, Model I -Two Ca$^{2+}$

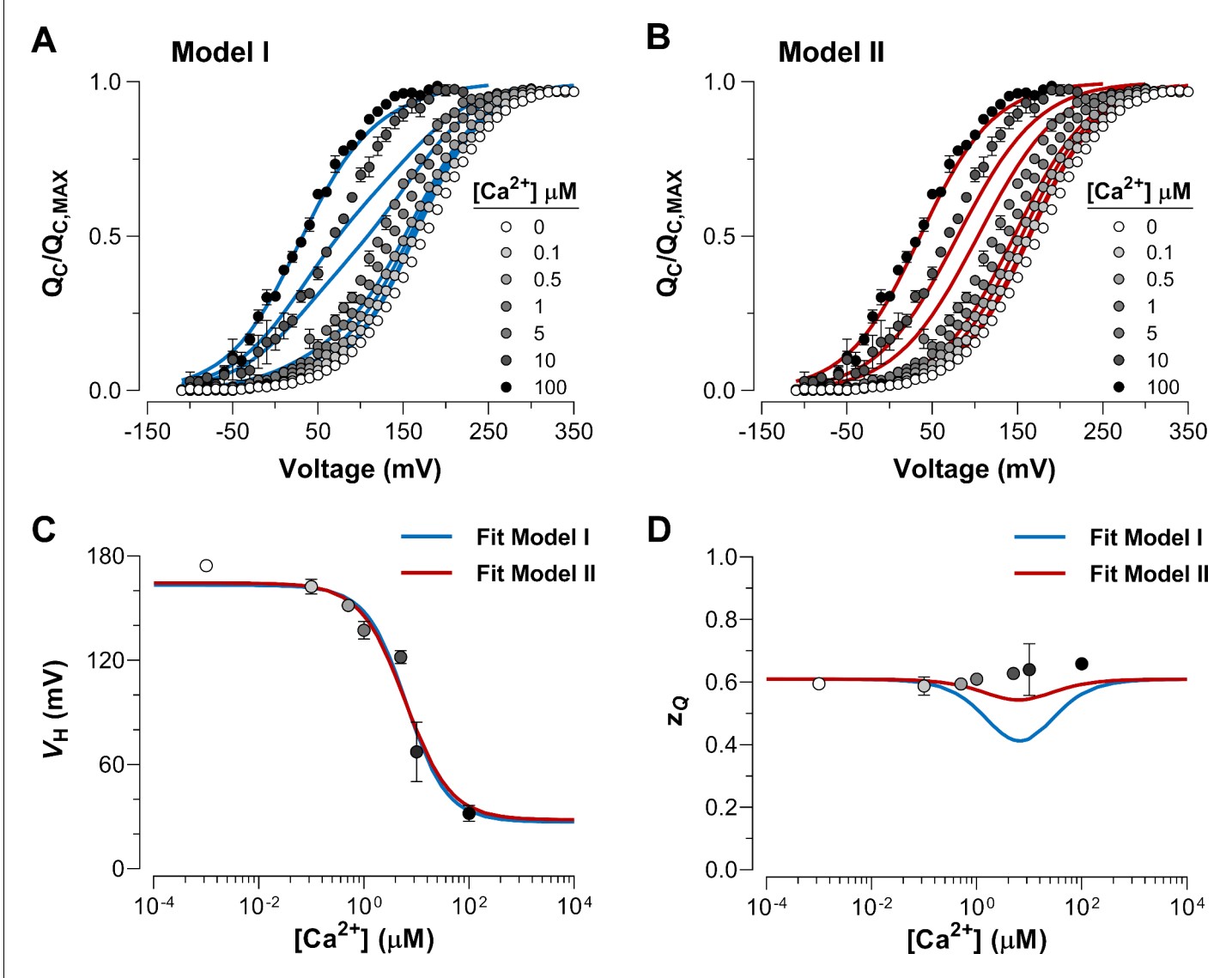

**Figure 4.** Dose-dependent effect of Ca$^{2+}$ on voltage sensor activation is predicted by a Ca$^{2+}$-VSD interaction mechanism in which Ca$^{2+}$-binding affects equally the VSD in all four α-subunits. (A–B) The experimental $Q_C(V)$ data were fitted using the two possible allosteric interaction mechanisms between voltage and calcium sensors described by Scheme I and Scheme II. The blue and red lines represent the global fits by Model I and Model II, respectively. The allosteric factor $E$ ($E_{M1}$ and $E_{M2}$) was constrained to the value obtained from the individual fitting of the $Q_C(V)/Q_{C, MAX}$ curves at 0 and 100 μM Ca$^{2+}$ (experimental $E$ ($E_{exp}$) equal to 26.4, see *Table 1*). The $z_J$, $J_0$ and $K_D$ parameters were allowed to vary freely. Note that the allosteric factor $E$ for Model I ($E_{M1}$) and Model II ($E_{M2}$) have different interpretations, since $E_{M1} = E_{exp}$ whereas $E_{M2} = \sqrt[4]{E_{exp}}$ given that the four voltage sensor will be altered in 2.3-fold ($E_{M2} = 2.27$) with each additional Ca$^{2+}$ bound. (C–D) The Ca$^{2+}$-dependence of $z_Q$-$Q_C(V)$ (C) and $V_H$-$Q_C(V)$ (D) curves are superimposed with the $z_Q$ and $V_H$ values predicted by Model I (blue line) and Model II (red line).

DOI: https://doi.org/10.7554/eLife.44934.009

The following figure supplement is available for figure 4:

**Figure supplement 1.** Fitting of experimental $Q_C(V)$ data by the Ca$^{2+}$-VSD interaction mechanisms including two Ca$^{2+}$-binding sites per α-subunit.

DOI: https://doi.org/10.7554/eLife.44934.010

sites predicts a prominent Ca$^{2+}$- dependence of the $z_Q$ whereas as we found experimentally, the $z_Q$ predicted by Modell II is essentially independent of the Ca$^{2+}$ concentration. The slight apparent increase in $z_Q$ data observed in *Figure 4—figure supplement 1C* as the Ca$^{2+}$ concentration is increased is not statistically significant.

**Table 1.** Parameters for the best fits of the $Q_C(V)$ data using different Ca²⁺-VSD interaction models.

| One Ca²⁺-site per α-subunit | | | Two Ca²⁺-sites per α-subunit | | | | |
| --- | --- | --- | --- | --- | --- | --- | --- |
| | | | | Model I | | Model II | |
| Parameters | Model I | Model II | Parameters | With cooperativity | Without cooperativity | With cooperativity | Without cooperativity |
| $z_J$ | 0.61 | 0.61 | $z_J$ | 0.61 | 0.61 | 0.61 | 0.61 |
| $J_0$ | 0.020 | 0.018 | $J_0$ | 0.019 | 0.019 | 0.021 | 0.019 |
| $E$ | 26.4* | 2.27* | $E_{S1}$ | 4.57* | | 1.46* | |
| | | | $E_{S2}$ | 5.35* | | 1.52* | |
| $K_D$ (µM) | 6.4 | 6.1 | $K_{D1}$ (µM) | 3.2 | 4.9 | 837.7 | 5.9 |
| | | | $K_{D2}$ (µM) | 631.7 | 6.9 | 6.6 | 5.9 |
| | | | $G$ | 56.1 | 1* | 120.4 | 1* |
| AIC | −948.4 | −1150.9 | | −1088.9 | −1090.3 | −1162.0 | −1147.5 |
| $L_i$ | $4^*10^{-47}$ | 0.004 | | $1^*10^{-16}$ | $3^*10^{-16}$ | 1 | $7^*10^{-4}$ |
| $w_i$ | $4^*10^{-47}$ | 0.004 | | $1^*10^{-16}$ | $3^*10^{-16}$ | 0.995 | 0.001 |

*Fixed parameters in the model fitting. AIC values correspond to Akaike Information Criterion to select the best fit model. $\mathcal{L}_i$ and $w_i$ are the relative likelihood and the weight of each model within the set of candidate models.

DOI: https://doi.org/10.7554/eLife.44934.013

We note here that when the intrasubunit cooperativity factor ($G$) is allowed to vary freely, Model II produces estimates of the $K_D$ parameters that are out the range relative to the apparent Ca²⁺ affinities previously reported in the literature for the Ca²⁺-binding sites ($K_{D(\text{RCK1})}$ = 13 - 24 µM and $K_{D(\text{RCK2})}$ = 3 - 5 µM) (*Sweet and Cox, 2008*; *Bao et al., 2002*; *Xia et al., 2002*). In addition, Model II reaches better estimates of the $K_D$ parameters (*Table 1*) if we consider non cooperative interactions between the Ca²⁺-binding sites. However, although the $K_D$ value for the RCK2 Ca²⁺-binding site is in agreement with previous reports, Model II underestimates the value for $K_{D(\text{RCK1})}$.

## Discussion

Recent functional and structural studies have revealed the existence of a major interplay between voltage- and Ca²⁺-sensing modules in the BK channel (*Yuan et al., 2010*; *Savalli et al., 2012*; *Miranda et al., 2013*; *Miranda et al., 2016*; *Miranda et al., 2018*; *Carrasquel-Ursulaez et al., 2015*; *Hite et al., 2017*; *Tao et al., 2017*; *Zhang et al., 2017*), which offer a new perspective on our understanding of its multimodal gating mechanism. However, the details of the CTD-VSD allosteric coupling as well its molecular nature have yet to be firmly established since their direct assessment is experimentally challenging. Based on the functional independence of the distinct structural domains involved (PD, CTD, and VSD), the energetic relationship between the sensory modules can be directly defined by comparing the change in the equilibrium of the voltage sensor under two extreme Ca²⁺ conditions: unliganded and saturated (*Horrigan and Aldrich, 2002*).

Thus, by characterizing the voltage dependence of charge movement in the virtual absence of internal Ca²⁺ and at Ca²⁺ concentrations that saturate the Ca²⁺ high-affinity sites, this work directly establishes that Ca²⁺-binding significantly facilitates VSD activation through direct energetic contribution to the R-A equilibrium ($\Delta V_H$ = -142.6 ± 4.5 mV and $\Delta\Delta G_V^{Ca}$ = -7.98 ± 0.27 kJ/mol). Although this result is in agreement with previous reports from our laboratory ($\Delta V_H$ = -140 mV and $\Delta\Delta G_V^{Ca}$ = -7.9 kJ/mol; at [Ca²⁺]ᵢ = 100 µM) (*Carrasquel-Ursulaez et al., 2015*), it is at odds with the smaller leftward shift obtained by *Horrigan and Aldrich (2002)* at saturating Ca²⁺ concentration ($\Delta V_H$ = -33 mV and $\Delta\Delta G_V^{Ca}$ = -1.9 kJ/mol; at [Ca²⁺]ᵢ = 70 µM). The reason for the contradictory findings is not clear to us. At this stage, we can only identify two differences between the experimental procedures followed by *Horrigan and Aldrich (2002)* and by us. First, there is a difference in the BK channel clones used (*hSlo1* vs *mSlo1*). Since mBK channels share a high degree of identity with hBKs (96%), it seems unlikely that the differences in Ca²⁺ binding and voltage sensor coupling are due to different BK clones. Second, Horrigan and Aldrich did all their experiments at 0 and 70 µM Ca²⁺ in HEK cells. Here, one can argue that the strength of coupling between Ca²⁺ binding and the voltage sensor

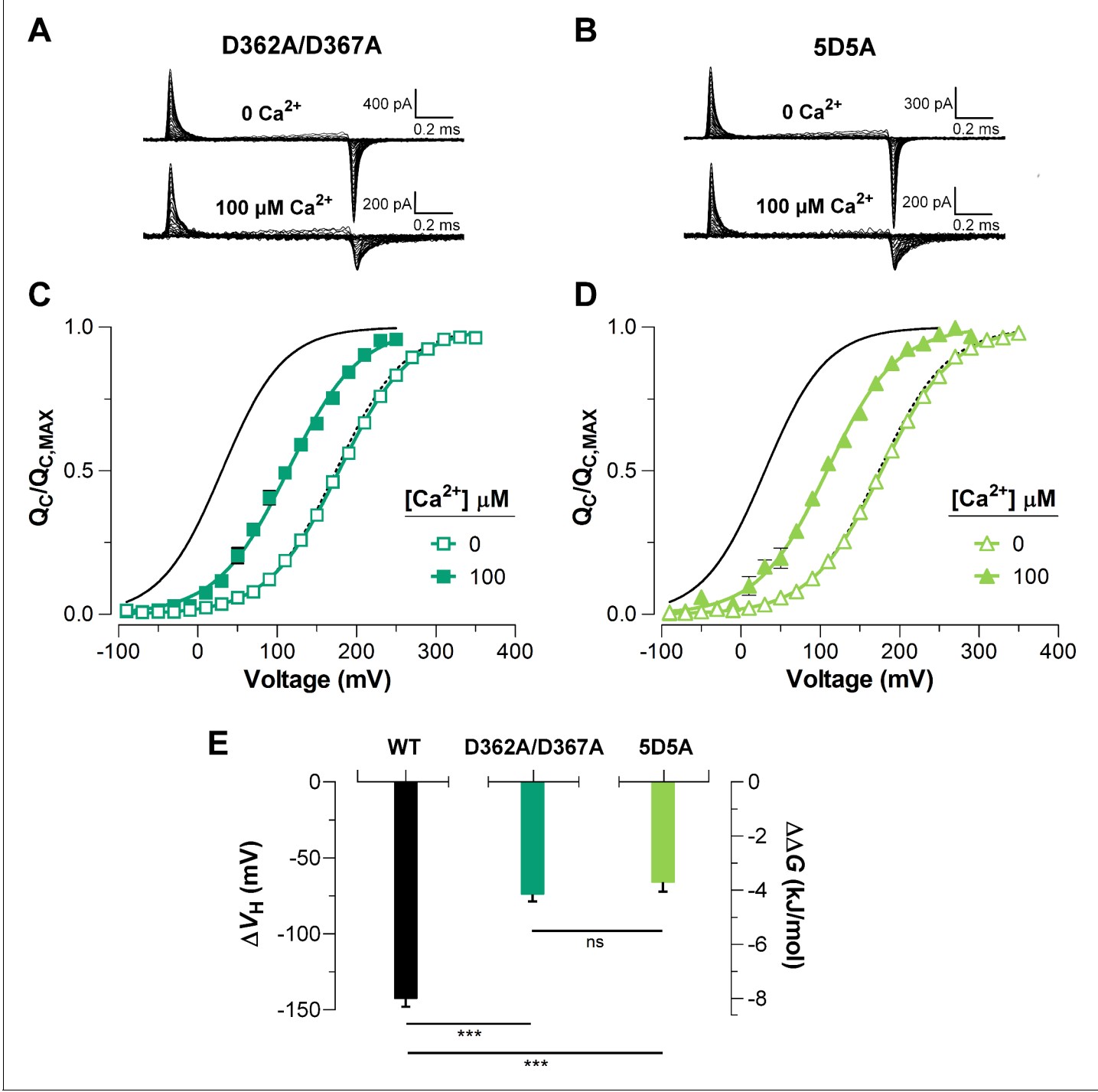

**Figure 5.** The high-affinity $Ca^{2+}$-binding sites contribute equally to the allosteric coupling between calcium and voltage sensors in BK channels. (**A–B**) Representative gating current ($I_G$) recordings at 0 and 100 µM of $[Ca^{2+}]_i$ for the RCK1 site mutant (D362A/D367A) and the RCK2 site mutant (5D5A), respectively. (**C–D**) Gating charge-voltage curves ($Q_C(V)$) were obtained at 0 $Ca^{2+}$ (open symbols) and 100 µM $Ca^{2+}$ (filled symbols) for D362A/D367A and 5D5A mutants, respectively. Boltzmann fitting to the experimental data (mean ± SEM) is indicated by solid lines ($V_{H(D362A/D367A)}$ = 178.0 ± 2.7 mV, $z_Q$ = 0.58 ± 0.01, $n$ = 12 and $V_{H(5D5A)}$ = 176.4 ± 4.6 mV, $z_Q$ = 0.58 ± 0.01, $n$ = 17 at 'zero' $Ca^{2+}$; $V_{H(D362A/D367A)}$ = 104.2 ± 7.3 mV, $z_Q$ = 0.56 ± 0.02, $n$ = 7 and $V_{H(5D5A)}$ = 110.8 ± 6.7 mV, $z_Q$ = 0.58 ± 0.02, $n$ = 6 at 100 µM $Ca^{2+}$). For comparison, all $Q_C(V)$ plots include the Boltzmann fit of the $Q_C(V)$ curves for WT at 0 $Ca^{2+}$ (dashed black line) and 100 µM $Ca^{2+}$ (solid black line). (**E**) Quantification of the $V_H$ shift ($\Delta V_H$) in the $Q_C(V)$ curves and the free energy change ($\Delta\Delta G_V^{Ca}$) induced by 100 µM $Ca^{2+}$. The non-parametric $t$-test was used to evaluate statistical significances between WT BK channel and the RCK sites mutants (***p<0.001).

DOI: https://doi.org/10.7554/eLife.44934.011

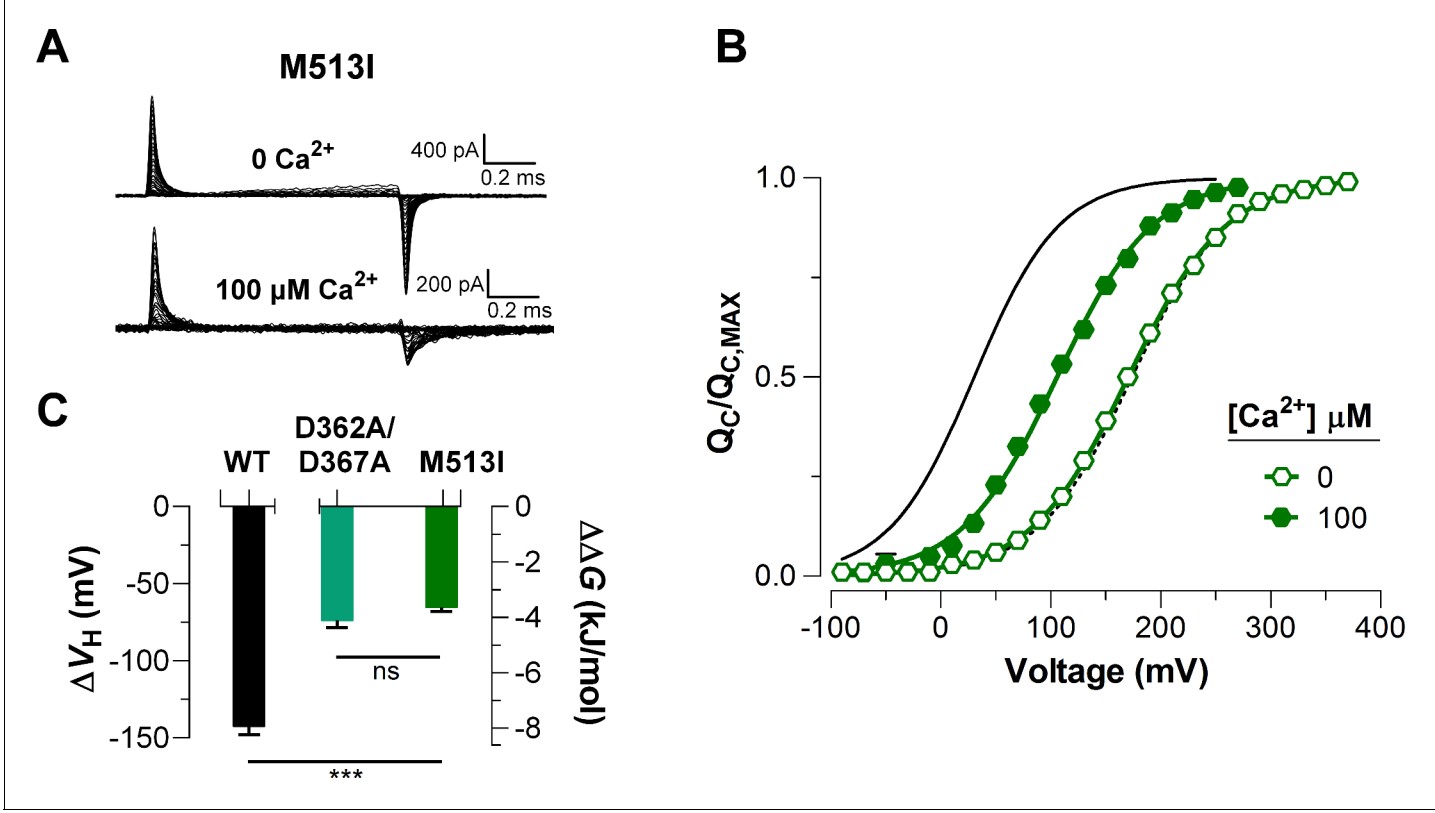

**Figure 6.** Mutations abolishing Ca$^{2+}$-sensing by the RCK1 binding-site reduce the Ca$^{2+}$-induced effect on voltage sensors activation similarly. (**A**) Representative gating current (I$_G$) recordings at 0 and 100 µM [Ca$^{2+}$]$_i$ for the RCK1 site mutant M513I. (**B**) Gating charge-voltage curves $Q_C(V)$ were obtained at 0 Ca$^{2+}$ and 100 µM Ca$^{2+}$ (open and filled symbols) for the M513I mutant. Boltzmann fit to the experimental data (mean ± SEM) is indicated by solid lines ($V_{H(M513I)}$= 170.4 ± 4.4 mV, $z_Q$ = 0.58 ± 0.01, $n$ = 17 at 'zero' Ca$^{2+}$ and $V_{H(M513I)}$= 105.0 ± 6.3 mV, $z_Q$ = 0.62 ± 0.03, $n$ = 4 at 100 µM Ca$^{2+}$). For comparison, the $Q_C(V)$ plot includes the Boltzmann fit of the $Q_C(V)$ curves for WT at 0 Ca$^{2+}$ and 100 µM Ca$^{2+}$ (dashed and solid black line). (**C**) Quantification of the $V_H$ shift ($\Delta V_H$) in the $Q_C(V)$ curves and the free energy change induced by 100 µM Ca$^{2+}$ ($\Delta\Delta G_V^{Ca}$). A non-parametric t-test was used to compare WT and RCK1-site mutants BK channels (***$p<0.001$).
DOI: https://doi.org/10.7554/eLife.44934.012

may be different in distinct expression systems (e.g., differential modulation) but this is something that is outside the scope of the present paper. Moreover, even if we assume that the calcium effect on VSD is underestimated at 70 µM Ca$^{2+}$ (*Horrigan and Aldrich, 2002*) compared to 100 µM (our work), we observed a significantly greater effect of Ca$^{2+}$ concentrations (1, 5 and 10 µM, *Figure 2*) when less than 50% of the Ca$^{2+}$ sensors are occupied ($K_D$ = 11 µM; *Horrigan and Aldrich, 2002*; *Cox et al., 1997*).

Fluorescence studies that optically track the motion of the voltage sensor or of the gating ring provide two lines of evidence that support the present findings. First, conformational rearrangements of the voltage sensors detected using voltage-clamp fluorometry can be provoked by Ca$^{2+}$-binding to the high-affinity sites. A sudden rise in intracellular [Ca$^{2+}$] caused by a UV flash induced-photolysis of caged Ca$^{2+}$ prompts a leftward shift in both conductance-voltage ($G(V)$) and fluorescence-voltage ($F(V)$) relationships. These results suggest that functional activation of the gating ring is propagated to the VSD, leading to structural perturbations of voltages sensors, thereby favoring its active conformation (*Savalli et al., 2012*). Second, the structural rearrangement of the gating ring in response to Ca$^{2+}$ has a voltage dependence (*Miranda et al., 2013*; *Miranda et al., 2018*) attributable to the operation of the voltage sensor. The origin of these voltage-dependent motions has recently been established via modifications of the voltage-sensing function of the BK channel using the patch-clamp fluorometry technique (*Miranda et al., 2018*). Both mutations of the charged residue on the S4 transmembrane segment (R210, R213, and E219) and the co-expression of the β1-subunit with BKα channel, modify the conformational changes of the gating ring triggered by

depolarization in correspondence to the observed $G(V)$ shift measured for these channel constructs. In contrast, perturbations of pore opening equilibrium (e.g. through the F315A mutation or the assembly of BKα channel with γ1-subunit) does not modify the voltage-dependent reorganization of the gating ring (*Miranda et al., 2018*).

Mechanistically, how might the CTD-VSD coupling occur in manner that is independent of channel opening? Taking into account the homotetrameric configuration of the BK channel, *Horrigan and Aldrich (2002)* defined the general gating scheme of BK channel considering the simplest CTD-VSD interaction model in which voltage sensors and Ca$^{2+}$-binding sites interact solely within the same subunit. However, the VSD movement at non-saturating Ca$^{2+}$ conditions observed here, which entail distinct functional states of the Ca$^{2+}$-binding sites (unliganded and liganded), unveiled that the standard HA model is unable to explain the mechanistic interaction governing the allosteric coupling between the Ca$^{2+}$ and voltage sensors. Assuming that Ca$^{2+}$-sensors are independent and they modify the voltage sensor in the same subunit only, Scheme I would predict $Q_C(V)$ curves characteristic of an all-or-none model showing two well-distinguishable Boltzmann components corresponding to the fractions of unliganded and Ca$^{2+}$-saturated BK channels (*Figure 4A*). Conversely, an energetic effect of each Ca$^{2+}$-site on all the voltage sensors of the tetramer would lead to an equivalent functional status of each VSD, so that the $Q_C(V)$ curves would shift as the occupancy of the Ca$^{2+}$ sites increased. Proposing that the VSD and Ca$^{2+}$ sites interact in the manner described by Scheme II reproduces reasonably well the behavior of the Ca$^{2+}$-dependent gating charge movement observed in our experiments (*Figure 4B–D*). This concerted CTD-VSD communication may underlie a mechanism analogous to the mechanical strategy of interaction between the homooctameric ring of RCK domains and the pore module described for bacterial K$^+$ channels (*Jiang et al., 2002*; *Ye et al., 2006*; *Lingle, 2007*; *Pau et al., 2011*; *Smith et al., 2012*; *Smith et al., 2013*). In both MthK and BK channels the Ca$^{2+}$-site occupancy triggers a conformational change corresponding to a symmetric overall rearrangement of the cytosolic tetrameric structure that is ultimately propagated to the transmembrane regions (TMD) via C-linker and in the BK channel, also via the protein-protein interfaces between the gating ring and the TMD (*Jiang et al., 2002*; *Jiang et al., 2003*; *Ye et al., 2006*; *Yuan et al., 2010*; *Yuan et al., 2012*; *Pau et al., 2011*; *Smith et al., 2012*; *Tao et al., 2017*). Thus, we can speculate that each Ca$^{2+}$-binding event produces a gradual conformational expansion of the gating ring affecting the four voltage sensors in each step through the progressive perturbations within the protein-protein interfaces.

As mentioned above, the communication pathway through which the Ca$^{2+}$-driven conformational changes are propagated to the voltage sensors appears to reside on the CTD-VSD interface that involves non-covalent interactions between RCK1 N-lobe and S0-S4 transmembrane segments (*Yang et al., 2007*; *Yang et al., 2008*; *Yang et al., 2010*; *Sun et al., 2013*; *Hite et al., 2017*; *Tao et al., 2017*). Although it is not possible at present to dismiss the possibility that the Ca$^{2+}$ binding effect on the VSD workings is, at least in part, mediated by the covalent pulling of the C-linker, we recall that the VSDs are domain-swapped with RCK domains in the gating ring (*Tao et al., 2017*). Therefore, binding of Ca$^{2+}$ makes the RCK1 N-lobe pull on the S6 helix from its subunit whereas the modification of the contact surface between the RCK1 N-lobe with the voltage sensor of an adjacent subunit induces an outward displacement of the voltage sensor (*Hite et al., 2017*). These structural arguments make us favor the non-covalent interaction between the CTD and the VSD as the source of the coupling between these two structures. Scanning mutagenesis of RCK1-N terminal subdomain indicates that residues on the βA-αC region are involved in the allosteric connection of the Ca$^{2+}$-dependent activation mediated by RCK1 site occupancy (*Yang et al., 2010*). In line with this study, the selective activation of the RCK1 domain was identified as being responsible for the Ca$^{2+}$-induced VSD rearrangement (*Savalli et al., 2012*) and the voltage dependence of the Ca$^{2+}$-driven motions of gating ring (*Miranda et al., 2016*; *Miranda et al., 2018*), suggesting that CTD-VSD allosteric coupling is primarily determined by the RCK1 site (*Sweet and Cox, 2008*). However, our results are inconsistent with this picture. The constructs D362A/D367A and 5D5A (D894A-D898A) selectively impair the Ca$^{2+}$-sensitivity of the RCK1- and RCK2-sensors, respectively, by neutralizing the residues that are involved in contributing to Ca$^{2+}$-coordination (*Zhang et al., 2010*; *Tao et al., 2017*). Comparing the fast gating charge movement at 0 Ca$^{2+}$ and saturating Ca$^{2+}$ conditions reveals that the energetic effect of Ca$^{2+}$-binding on voltage sensor equilibrium is practically identical ($\sim -4$ kJ/mol) for both the D362A/D367A and the 5D5A mutations (*Figure 5*). Thus, our findings establish that the RCK2-driven contribution to CTD-VSD energetic coupling is quite similar to the RCK1-driven

contribution. The functional role of the RCK2-sensor on $Ca^{2+}$-sensitivity of VSD activation was further corroborated using the M513I mutation (*Figure 6*). This point mutation hinders the $Ca^{2+}$-dependent activation associated with the RCK1-sensor, presumably by disrupting the structural integrity of the binding site and the transduction pathway through the βA-αC region (*Zhang et al., 2010*). Thus, another residue involved in the BK $Ca^{2+}$-dependent activation mediated by the RCK1 $Ca^{2+}$-binding site but not forming part of the site itself, decreases the $Q_C(V)$ leftward shift almost in the same amount as does the D362A/D367A mutant.

We found that the energetic contribution of each RCK site to the voltage sensor equilibrium is the same and its combination mimics the VSD $Ca^{2+}$-sensitivity of the fully occupied sites. These findings remind us of early reports showing that mutations in each RCK site shift the $Ca^{2+}$-dependent $G(V)$ by approximately one-half relative to the effect seen for WT channels (*Bao et al., 2002*; *Xia et al., 2002*). Thus, our results suggest that the two RCK-sensors contribute independently to the modulation on the VSD, although we cannot eliminate the possibility of some cooperativity between them. Indeed, various lines of evidence indicate that there is some, albeit modest, cooperativity between the two high-affinity $Ca^{2+}$-binding sites although its nature is still unclear (*Qian et al., 2006*; *Savalli et al., 2012*; *Sweet and Cox, 2008*). Intra and intersubunit structural connectivity supports the putative cooperative interactions between the $Ca^{2+}$ sensors at the gating ring (*Hite et al., 2017*; *Yuan et al., 2012*). In fact, a recent functional study of the intrasubunit connections between the RCK1 site and $Ca^{2+}$ bowl (R514-Y904/E902 interactions) has shown that such connections are potential candidates for the structural determinants underlying a cooperative mechanism between the RCK1- and RCK2-sensor. These interactions are involved in either the preservation of the integrity of RCK1 $Ca^{2+}$-binding site or define the allosteric propagation pathway of the chemical energy induced by $Ca^{2+}$ binding towards transmembrane domains (*Kshatri et al., 2018*). On the basis of the cryo-EM structure of *Aplysia californica* BK channel, (*Hite et al., 2017*) proposed that there should be a positive cooperativity between the $Ca^{2+}$-binding at RCK1 site and the $Ca^{2+}$ bowl since the $Ca^{2+}$-induced conformational change of the RCK1-N lobes from closed to open configuration depends on the functional state (unliganded and liganded) of both RCK sites.

Our analysis based on the CTD-VSD interaction model can not specify likely cooperative relations among the two high-affinity $Ca^{2+}$ sites within the same α-subunit. In fact, the analysis of our gating current data when using cooperativity between the two high-affinity sites (*Table 1*) produced a result at odds with previous reports. Based on the structural information of the BK channels, it could be considered that cooperative interactions between the $Ca^{2+}$ sensors of the different α-subunits (*Hite et al., 2017*) can also account, in part, for the $Ca^{2+}$-dependent behavior of the VSDs. However, functional studies point out a more relevant role of the intrasubunit cooperativity (albeit modest) than the intersubunit cooperativity between the RCK $Ca^{2+}$ sites (*Niu and Magleby, 2002*; *Qian et al., 2006*). Thus, although a concerted CTD-VSD model (Scheme II) gives better explanation than an independent CTD-VSD model (Scheme I) to the allosteric communication of the calcium and voltage sensors, more work is required to explore improved models able to reproduce more accurately the properties of the interaction CTD-VSD mechanism in BK channel.

In conclusion, our results depict a remarkable, and direct energetic interplay between the specialized sensory modules (VSD and CTD) of the BK channel. Our findings together with the emerging structural-functional information establish a new paradigm about how stimuli integration (depolarization and intracellular $Ca^{2+}$) modulates this channel's activation and its relevance within a physiological context. Notable and unexpected is the equivalent contribution of the distinct ligand-binding sites in the cytosolic domain to the allosteric regulation of voltage sensing. Additional studies to discern the molecular bases underlying the $Ca^{2+}$ and voltage propagation pathways and the cooperative interactions of the RCK1 and RCK2 regulatory domains may provide new clues about the dual gating mechanism of BK channel.

## Materials and methods

### Channel expression

*Xenopus laevis* oocytes were used as a heterologous system to express BK channels. The cDNA coding for the human BK α-subunit (U11058) was provided by L. Toro (University of California, Los Angeles, CA). The cDNA coding for independent mutants of each two high-affinity $Ca^{2+}$ site from

BK channel, the double mutant D362A/D367A (*Xia et al., 2002*), the mutant M513I (*Bao et al., 2002*) in the RCK1 $Ca^{2+}$-binding site, and the mutant 5D5A (*Schreiber and Salkoff, 1997*) (D894A/D895A/D896A/D897A/D898A) in the RCK2 $Ca^{2+}$-binding site or calcium bowl, were kindly provided by M. Holmgren (National Institutes of Health, Bethesda, MD). The cRNA was prepared using mMESSAGE mMACHINE (Ambion) for in vitro transcription. *Xenopus laevis* oocytes were injected with 50 ng of cRNA and incubated in an ND96 solution (in mM: 96 NaCl, 2 KCl, 1.8 $CaCl_2$, 1 $MgCl_2$, 5 HEPES, pH 7.4) at 18°C for 4–8 days before electrophysiological recordings.

## Electrophysiological recordings

All recordings were made by using the patch-clamp technique in the inside-out configuration. Data were acquired with an Axopatch 200B (Molecular Devices) amplifier and the Clampex 10 (Molecular Devices) acquisition software. Gating current ($I_G$) records were elicited by 1 ms voltage steps from −90 to 350 mV in increments of 10 mV. Both the voltage command and current output were filtered at 20 kHz using an 8-pole Bessel low-pass filter (Frequency Devices). Current signals were sampled with a 16-bit A/D converter (Digidata 1550B; Molecular Devices), using a sampling rate of 500 kHz. Linear membrane capacitance and leak subtraction were performed based on a P/4 protocol (*Armstrong and Bezanilla, 1974*).

Borosilicate capillary glasses (1B150F-4, World Precision Instruments) were pulled in a horizontal pipette puller (Sutter Instruments). After fire-polishing, pipette resistance was 0.5–1 MΩ. The external (pipette) solution contained (in mM): 110 tetraethylammonium (TEA)-$MeSO_3$, 10 HEPES, 2 $MgCl_2$; pH was adjusted to 7.0. The internal solution (bath) contained (in mM): N-methyl-D-glucamine (NMDG)-$MeSO_3$, 10 HEPES, and 5 EGTA for 'zero $Ca^{2+}$' solution (~0.8 nM, based on the presence of ~10 μM contaminant [$Ca^{2+}$] (*Cui et al., 1997*). An agar bridge containing 1 M NaMES connected the internal solution to a pool of the external solution grounded with an Ag/AgCl electrode. The calculated bridge/bath junction potential was ~0.8 mV. For test solutions at different $Ca^{2+}$ concentrations (0.1–100 μM), $CaCl_2$ was added to reach the desired free [$Ca^{2+}$], and 5 mM EGTA (0.1–0.5 μM) or HEDTA (1–10 μM) was used as calcium buffer. No $Ca^{2+}$ chelator was used in 100 μM $Ca^{2+}$ solutions. Free calcium concentration was estimated using the WinMaxChelator Software and checked with a $Ca^{2+}$-electrode (Hanna Instruments). All experiments were performed at room temperature (20–22°C). To measure $I_G$ at different $Ca^{2+}$ concentrations in the same oocyte, the patch was excised and washed with an appropriate internal solution using at least 10 times the chamber volume.

## Data analysis

All data analysis was performed using Clampfit 10 (Molecular Devices, RRID:SCR_011323), Matlab (MathWorks, RRID:SCR_001622) and Excel 2007 (Microsoft, RRID:SCR_016137). The first 50-100 μs of the ON-gating currents were fitted to a single exponential function and the area under the curve (*Figure 1A-B*) was integrated to obtain the charge displaced between closed states ($Q_C$) (*Horrigan and Aldrich, 1999*; *Horrigan and Aldrich, 2002*; *Carrasquel-Ursulaez et al., 2015*; *Contreras et al., 2012*). $Q_C(V)$ data for each [$Ca^{2+}$]$_i$ were fitted using a Boltzmann function:

$$Q_C(V) = \frac{Q_{C,\,MAX}}{1 + e^{\left(\frac{-z_Q F(V - V_H)}{RT}\right)}}$$

where $Q_{C,\,MAX}$ is the maximum charge, $z_Q$ is the voltage dependence of activation, $V_H$ is the half-activation voltage, $T$ is the absolute temperature (typically 295 K), $F$ is the Faraday's constant, and $R$ is the universal gas constant. $Q_{C,\,MAX}$, $V_H$, and $z_Q$ were determined using least square minimization. $Q_C(V)$ curves were aligned by shifting them along the voltage axis by the mean $\Delta V = (\langle V_H \rangle - V_H)$ to generate a mean curve that did not alter the voltage dependence (*Horrigan et al., 1999*). All error estimates are SEM.

For each experiment, the time constants obtained from exponential fits to ON-gating currents were shifted along the voltage axis by $\Delta V$ to determine the mean $\tau_{I_G - ON}(V)$ relationships. $\tau_{I_G - ON}(V)$ data were fitted to a two-states process described by

$$\tau(V) = 1/(\alpha(V) + \beta(V))$$

where $\alpha(V) = \alpha_0 e^{(z_J \delta F V / RT)}$ and $\beta(V) = \beta_0 e^{(z_J (\delta - 1) F V / RT)}$ are, respectively, the forward and backward rate constants which determine the equilibrium constant $J$ of the voltage sensor (see below). The parameter $\delta$ is the electrical distance at which the peak of the energy barrier that separates the two resting (R)-active (A) states of the voltage sensor is located, and $z_J$ is the number of gating charges displaced during the R-A transition.

The $Ca^{2+}$-induced effect on VSD activation was quantified as the $V_H$ shift relative to 'zero' $Ca^{2+}$ condition: $\Delta V_H = V_H\left([Ca^{2+}]_i\right) - V_H\left(0\,[Ca^{2+}]_i\right)$. For wild-type (WT) BK channel and the RCK $Ca^{2+}$-sensor mutants (D362A/D367A, M513I and 5D5A), the energetic contribution of $Ca^{2+}$-binding on resting-active (R-A) equilibrium of the voltage sensor was calculated as changes in Gibbs free energy of VSD activation induced by 100 µM $Ca^{2+}$:

$$\Delta\Delta G_V^{Ca} = F \left( z_{Q\left(100\,\mu M\,[Ca^{2+}]_i\right)} V_{H\left(100\,\mu M\,[Ca^{2+}]_i\right)} - z_{Q\left(0\,[Ca^{2+}]_i\right)} V_{H\left(0\,[Ca^{2+}]_i\right)} \right)$$

## Model fitting

We fit the $Q_C(V, [Ca^{2+}])$ experimental data using two distinct interaction mechanisms between $Ca^{2+}$-binding sites and voltage sensor (see Scheme I and Scheme II in the *Figure 3A-B*) within the framework of Horrigan-Aldrich (HA) general allosteric model (*Horrigan and Aldrich, 2002*). Assumptions and considerations for the equations that describe each one of the $Ca^{2+}$-VSD interaction schemes are given in the *Appendix*. In terms of the HA allosteric mechanisms, the voltage sensor R-A equilibrium is defined by the equilibrium constant $J$ according to the relation:

$$J = e^{\frac{z_J F (V - V_H)}{RT}} = J_0 e^{\frac{z_J F V}{RT}}$$

where $J_0$ is the zero voltage equilibrium constant and $z_J$ the gating charge displacement per voltage sensor. Thus, the fraction of the total charge displaced essentially between closed states, $\left(Q_C(V) / Q_{C,\,MAX}\right)$ in the absence of calcium can be written as:

$$Q_C(V) / Q_{C,\,MAX}\left(Ca^{2+} \ll K_D\right) = \frac{1}{1 + J^{-1}}$$

where $K_D$ is the dissociation constant of the high-affinity calcium-binding site with all voltage sensors at rest and the channel closed. In the presence of saturating $Ca^{2+}$ (100 µM), the equilibrium of the R-A transition $J$ becomes amplified by the allosteric factor $E$, which defines the coupling between $Ca^{2+}$-binding sites and voltage sensors, being

$$Q_C(V) / Q_{C,\,MAX}\left(Ca^{2+} \gg K_D\right) = \frac{1}{1 + (JE)^{-1}}$$

and

$$JE = J_0 e^{\frac{RT \ln(E) + z_J F V}{RT}}$$

The $Q_C(V) / Q_{C,\,MAX}$ measured in the presence of high $[Ca^{2+}]$ and 'zero $Ca^{2+}$' condition at the same voltage (so that $J$ be canceled out) but in the limit where $(JE)^{-1} \gg 1$ is:

$$\frac{Q_C(V) / Q_{C,\,MAX}\left(Ca^{2+} \gg K_D\right)}{Q_C(V) / Q_{C,\,MAX}\left(Ca^{2+} \ll K_D\right)}_{\left(\lim JE^{-1} \gg 1\right)} = \frac{JE}{J} = E$$

Thus, the Gibbs free energy perturbation of the voltage sensor R-A equilibrium when the high-affinity binding sites are approximately 100% occupied by $Ca^{2+}$ (100 µM) is a straightforward measure of the allosteric factor: $E = e^{-\Delta\Delta G_V^{Ca} / RT}$.

Based on these conditions, the values of the allosteric parameter $E$ were calculated and introduced in each of the two $Ca^{2+}$-VSD interaction models as a fixed parameter. Once $E$ was obtained, the families of $Q_C(V, [Ca^{2+}])$ curves were simultaneously fitted to *Equation 4* and *Equation 6* (see Appendix) and estimating the $z_J$, $J_0$ and $K_D$ parameters for each model by minimizing least-square values. To select the best $Ca^{2+}$-VSD interaction scheme that describes the experimental data, the fits provided by each model were compared according to their Akaike Information Criterion (AIC)

values (*Akaike, 1974*), calculated as $\text{AIC}_i = 2p_i - 2\ln(L_i)$, where $p_i$ is the number of free parameters and $\ln(L_i)$ is the maximum log-likelihood of the model $i$. The best fit being the one that achieves the lowest $\text{AIC}_i$ value. Minimum $\text{AIC}_i$ ($\text{AIC}_{\text{MIN}}$) values were used as model selection criteria. Using the $\text{AIC}_i$ weights ($w_i$), we estimated the probability of model $i$ is the best model given the data and the set of candidate models. $w_i$ are based on the relative likelihood of each tested model $i$ which is a function of the difference in $\text{AIC}_i$ score and the best model: $\Delta\text{AIC}_i = \text{AIC}_i - \text{AIC}_{\text{MIN}}$ (*Anderson and Burnham, 2002*). From the $\Delta\text{AIC}_i$ we obtained an estimate of the relative likelihood of model $i$ ($\mathfrak{L}_i$) by the simple transform: $\mathfrak{L}_i = exp\left(-\frac{1}{2}\Delta\text{AIC}_i\right)$. $w_i$ is calculated normalizing the $\mathfrak{L}_i$ for each

model: $w_i = \mathfrak{L}_i / \sum\limits_{k=1}^{K} \mathfrak{L}_k$, where $K$ is the number of candidate models.

The models for the Ca$^{2+}$-VSD interaction schemes were extended including two high-affinity Ca$^{2+}$-binding sites per α-subunit (*Figure 3—figure supplement 1D-E*). The contribution of each Ca$^{2+}$-binding site to the free energy of the voltage sensor equilibrium may be split in two, such as $E = E_{S1} * E_{S2} = e^{-\left(\Delta\Delta G_V^{Ca}(S1) + \Delta\Delta G_V^{Ca}(S2)\right)/RT}$, where $E_{S1}$ and $E_{S2}$ are the allosteric factors $E$ for the RCK1 and RCK2 sites. Thus, for the global fit of the $Q_C(V, [\text{Ca}^{2+}])$ curves, we constrained the allosteric parameter $E_{S1}$ and $E_{S2}$ obtained experimentally for the RCK2 Ca$^{2+}$-sensor mutant (5D5A) and RCK1 Ca$^{2+}$-sensor mutant (D362A/D367A), respectively, as described above. The rest of the parameters $z_J$, $J_0$, $K_{D1}$, $K_{D2}$, and $G$, where $K_{D1}$ and $K_{D2}$ are the dissociation constants of the RCK1 and RCK2 sites and $G$ is a cooperativity factor between the two sites within the same α-subunit of the BK channel, were allowed to vary freely.

# Acknowledgements

We thank Dr. John Ewer for his comments on the manuscript and to Mrs. Luisa Soto (University of Valparaiso) for excellent technical assistance. This research was supported by FONDECYT Grant No. 1150273, FONDECYT Grant No. 1190203 and AFOSR No. FA9550-16-1-0384 to RL; CONICYT-PFCHA Doctoral fellowships No. 63140149 to YLC; FONDECYT Grant No. 1180999 to KC The Centro Interdisciplinario de Neurociencia de Valparaiso is a Millennium Institute supported by the Millennium Scientific Initiative of the Chilean Ministry of Economy, Development, and Tourism (P029-022-F).

# Additional information

## Funding

| Funder | Grant reference number | Author |
| --- | --- | --- |
| AFOSR | No. FA9550-16-1-0384 | Ramon Latorre |
| FONDECYT | Grant No. 1180999 | Karen Castillo |
| CONICYT-PFCHA | Doctoral fellowship No. 63140149 | Yenisleidy Lorenzo-Ceballos |
| Chilean Ministry of Economy, Development, and Tourism | Millennium Scientific Initiative P029-022-F | Yenisleidy Lorenzo-Ceballos Willy Carrasquel-Ursulaez Karen Castillo Osvaldo Alvarez Ramon Latorre |
| FONDECYT | Grant No. 1150273 | Ramon Latorre |
| FONDECYT | Grant No. 1190203 | Ramon Latorre |

The funders had no role in study design, data collection and interpretation, or the decision to submit the work for publication.

## Author contributions

Yenisleidy Lorenzo-Ceballos, Conceptualization, Formal analysis, Validation, Investigation, Visualization, Methodology, Writing—original draft; Willy Carrasquel-Ursulaez, Conceptualization, Software, Formal analysis, Investigation, Writing—review and editing; Karen Castillo, Validation,

Investigation, Methodology, Writing—review and editing; Osvaldo Alvarez, Conceptualization, Software, Formal analysis, Writing—review and editing; Ramon Latorre, Conceptualization, Resources, Data curation, Software, Formal analysis, Supervision, Funding acquisition, Validation, Investigation, Visualization, Methodology, Writing—original draft, Project administration, Writing—review and editing

### Author ORCIDs
Yenisleidy Lorenzo-Ceballos [iD] https://orcid.org/0000-0003-4309-9314
Ramon Latorre [iD] https://orcid.org/0000-0002-6044-5795

### Ethics
Animal experimentation: This study was performed in strict accordance with the recommendations in the Guide for the Care and Use of Laboratory Animals of the Ethics Committee for Animal Experimentation of the University of Valparaíso. All of the animals were handled according to approved institutional animal care and use committee protocols (BEA031-14) of the University of Valparaiso. All surgery was performed under tricaine anesthesia, and every effort was made to minimize suffering.

### Decision letter and Author response
Decision letter https://doi.org/10.7554/eLife.44934.017
Author response https://doi.org/10.7554/eLife.44934.018

## Additional files

### Supplementary files
• Transparent reporting form
DOI: https://doi.org/10.7554/eLife.44934.014

### Data availability
All data generated or analysed during this study are included in the manuscript and supporting files.

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

## Appendix 1

DOI: https://doi.org/10.7554/eLife.44934.015

### Assumptions and model testing

We assumed that the four voltage sensors act independently, transiting between two states, resting (R) and active (A), governed by the voltage-dependent equilibrium constant $J$. The R–A equilibrium is displaced toward the active state by membrane depolarization, generating a fast gating charge movement ($Q_C$) before channel opening. Additionally, the $Ca^{2+}$-binding to high-affinity sites shifts the voltage sensor equilibrium toward its active configuration through an allosteric coupling described by the factor $E$ (**Figure 3—figure supplement 1A**). By assuming the simplified standard model for the BK channels (**Horrigan and Aldrich, 2002**), where each $\alpha$-subunit has a single $Ca^{2+}$-binding site, we established the possible states and their connections through which each voltage sensor transits in the presence of $Ca^{2+}$ (**Figure 3—figure supplement 1B–C**) following the CTD-VSD interaction mechanisms described by Scheme I and II (**Figure 3A–B**).

For Scheme I, in which $Ca^{2+}$-binding sites and voltage sensors interact within the same $\alpha$-subunit, the activation of each VSD can occur through the $R_0$-$A_0$ or $R_1$-$A_1$ transitions according to the functional state of the $Ca^{2+}$ site (unbound or $Ca^{2+}$ bound). The equilibrium of such transitions is governed by $J$ and $JE_{M1}$, respectively (**Figure 3—figure supplement 1B**). In the case of Scheme II, in which binding of $Ca^{2+}$ to a single $\alpha$-subunit affects all four voltage sensors equally, the R-A equilibrium for each VSD would be affected by the number of $Ca^{2+}$ bound in the channel (0-4) depicted in the model (Model II) as five possible R-A transitions. According to this model, the $J$ constant increase $E_{M2}$-fold for each $Ca^{2+}$ site occupied (**Figure 3—figure supplement 1C**). For both schemes, the horizontal transitions R-R and A-A represent the $Ca^{2+}$-binding equilibrium ($K$ or $KE$) when the VSD is in the resting and active conformation, respectively. The equilibrium constant $K$ is defined as the bound/unbound probability ratio for each $Ca^{2+}$-binding site and depends on $Ca^{2+}$ concentration ($[Ca^{2+}]$) and the $Ca^{2+}$ dissociation constant ($K_D$): $K = [Ca^{2+}]/K_D$.

Here, we assumed that the voltage sensor movement at ON-gating currents is in equilibrium relative with the binding of $Ca^{2+}$. The assumption is reasonable since the $Ca^{2+}$-binding rate constant estimated for BK channel is $1.8 \times 10^8$ $M^{-1}s^{-1}$ (**Hou et al., 2016**) implying that at 10 μM internal $Ca^{2+}$ the association time constant is 340 μs. Thus, $Ca^{2+}$ binding at this $Ca^{2+}$ concentration proceeds with a time constant about six-fold longer than does the movement of the voltage sensor (~60 μs). Based on this consideration, the R-A transitions in the models would be predominant transitions whose proportion would be determined by the $[Ca^{2+}]$ and $K_D$. Therefore, predictions of the $Q_C(V)$ curves at different $Ca^{2+}$ concentrations for Model I and Model II were based on a given fractional occupancy of $Ca^{2+}$ sites established by the probability of $Ca^{2+}$ bound ($b$) and unbound ($1 - b$) for each $Ca^{2+}$-sensor, and the energetic contribution to VSD equilibrium.

Simulations of the $Q_C(V)$ curves using the Scheme I (Model I) were obtained using the equation

$$\frac{Q_C(V)}{Q_{C,MAX}} = (1 - b)\left(\frac{1}{1 + J^{-1}}\right) + b\left(\frac{1}{1 + (JE_{M1})^{-1}}\right); \tag{1}$$

where

$$b = \frac{1}{1 + K^{-1}} = \frac{1}{1 + \frac{K_D}{[Ca^{2+}]}} = \frac{[Ca^{2+}]}{[Ca^{2+}] + K_D}; \tag{2}$$

and

$$J = J_0 e^{\frac{z_J FV}{RT}} \tag{3}$$

Substituting $b$ and $J$ into **Equation 1**, the $Ca^{2+}$-dependent voltage sensor activation for Model I is given by the equation

$$\frac{Q_C(V)}{Q_{C,MAX}} = \left(\frac{K_D}{[Ca^{2+}] + K_D}\right)\left(\frac{1}{1 + \frac{e^{\frac{-z_J FV}{RT}}}{J_0}}\right) + \left(\frac{[Ca^{2+}]}{[Ca^{2+}] + K_D}\right)\left(\frac{1}{1 + \frac{e^{\frac{-z_J FV}{RT}}}{J_0 E_{M1}}}\right) \tag{4}$$

Thus, the $Q_C(V)$ curves are determined by the proportion of two functional VSD populations with a distinctive effect (unliganded effect or $Ca^{2+}$-saturated effect). Consequently, the $Q_C(V)$ curves are represented by a weighted sum of two Boltzmann functions.

Meanwhile, for the concerted CTD-VSD interaction Scheme II (Model II), the $Q_C(V, [Ca^{2+}])$ curves would be determined using the general equation:

$$\frac{Q_C(V)}{Q_{C,MAX}} = \sum_{x=0}^{n}\binom{n}{x}(1 - b)^{n-x} b^x \left(\frac{1}{1 + \left(JE_{M2}^x\right)^{-1}}\right) \tag{5}$$

The expression in the first bracket represents the fraction of VSD belonging to a channel with $x$ (0 to 4) $Ca^{2+}$ bound, according to a binomial probability distribution. Thus, the $Q_C(V)$ curves result in a weighted sum of five distinct Boltzmann functions corresponding to the five possible R-A transitions (**Figure 3—figure supplement 1C**). By setting $n = 4$ because the tetrameric symmetry of the channels, and substituting $b$ and $J$ into the previous equation (**Equation 5**) we have

$$\frac{Q_C(V)}{Q_{C,MAX}} = \sum_{x=0}^{4}\binom{4}{x}\left(\frac{K_D}{[Ca^{2+}] + K_D}\right)^{4-x}\left(\frac{[Ca^{2+}]}{[Ca^{2+}] + K_D}\right)^x \left(\frac{1}{1 + \frac{e^{\frac{-z_J FV}{RT}}}{J_0 E_{M2}^x}}\right) \tag{6}$$

It should be noted that at limiting $Ca^{2+}$ conditions, both schemes become equivalent where the VSD activation is characterized by a single Boltzmann function. At zero $Ca^{2+}$, the $Q_C(V)$ curves are described by

$$\frac{Q_C(V)}{Q_{C,MAX}} = \left(\frac{1}{1 + \frac{e^{\frac{-z_J FV}{RT}}}{J_0}}\right),$$

whereas $Ca^{2+}$ saturating concentration $J$ is multiplied by the allosteric factor $E$, where $E = E_{M1} = E_{M2}^4$ depending on the model (Model I or Model II):

$$\frac{Q_C(V)}{Q_{C,MAX}} = \left(\frac{1}{1 + \frac{e^{\frac{-z_J FV}{RT}}}{J_0 E}}\right)$$

Given that each α-subunit has two $Ca^{2+}$-binding sites, we expanded the CTD-VSD interaction schemes (**Figure 3—figure supplement 1C**) considering the existence of two $Ca^{2+}$-binding sites (**Figure 3—figure supplement 1D-E**). The Model I and II include the energetic contribution of RCK1 and RCK2 $Ca^{2+}$-sites to the VSD activation. The factor $E = E_{S1} * E_{S2}$ where $E_{S1}$ and $E_{S1}$ are the allosteric coupling between the VSD and the RCK1 $Ca^{2+}$-site and RCK2 $Ca^{2+}$-site, respectively. The $K_1$ and $K_2$ constants define the bound/unbound transition for each RCK1 and RCK2 sites with $K_1 = [Ca^{2+}]/K_{D1}$ and $K_2 = [Ca^{2+}]/K_{D2}$. Assuming that the $Ca^{2+}$ sensors of distinct α-subunit do not interact, we only consider intrasubunit cooperativity between the RCK1 and RCK2 sites defined by the factor $G$. Thus, the occupancy of one RCK site will affect $Ca^{2+}$-binding equilibrium to the other RCK site in the α-subunit ($GK_1$ and $GK_2$) (**Figure 3—figure supplement 1E**). Note that for Model II the

equilibrium $J$ of the VSD increase $E_{S1}$-fold and $E_{S2}$-fold for each $Ca^{2+}$ bound to the RCK1 and RCK2 sites, respectively, reaching $JE_{S1}^{4}E_{S2}^{4}$ when the eight $Ca^{2+}$ sites are occupied.

