## [Decision Letter]

Thank you for submitting your article "Calcium-driven regulation of voltage-sensing domains in BK channels" for consideration by *eLife*. Your article has been reviewed by three peer reviewers, and the evaluation has been overseen by a Reviewing Editor and Richard Aldrich as the Senior Editor. The reviewers have opted to remain anonymous.

The reviewers have discussed the reviews with one another and the Reviewing Editor has drafted this decision to help you prepare a revised submission.

The reviewers are in agreement that this paper represents an important contribution pertinent to the issue of how ligand-binding in BK channels may couple to channel activation. There is consensus that the results mainly embodied in Figure 1 are important and perhaps appropriate for *eLife*. However, the reviews together raise multiple, substantive points that can largely be broken into two categories. First, there are aspects of the data and analysis which require further elaboration. Second, to the extent the results challenge previous results, a rigorous, scholarly assessment of potential reasons for differences between the present results and those of Horrigan and Aldrich is required.

Regarding the first point, details are given in the reviewers' comments, perhaps most clearly from reviewer #1, but all reviewers are in agreement on these issues. Here, some of the major concerns are summarized. The individual reviews elaborate on several important major issues. They are included below.

1) It would be helpful to elaborate more clearly on how Q_C_ is determined, perhaps with examples, particularly at high Ca^2+^. The concern is that Q_C_ may not be adequately distinguished from Q_SS_, which is known to shift substantially with increases in Ca^2+^. In this regard, it would also be valuable to have examples of the time course of current activation under identical conditions of higher Ca^2+^ to provide assurance that Q_C_ is not being affected by channel opening. These concerns are extensively developed particularly in the comments from reviewer #1.

2) All reviewers note the apparent reduction in gating current amplitude at higher Ca^2+^. Although this may have a trivial explanation, some comment on this point should be provided. What was the time between each test step in the overall gating current protocol at a given Ca^2+^? Is Q_C,MAX_ changing with Ca^2+^? Does Q_OFF_=Q_ON_, as expected? Is each example in Figure 1A from a separate patch?

3) The shift in the Q_C_ /V relationship might also be expected to be associated with changes in kinetics, but this analysis was not done.

4) The attempt to test various models by which Ca^2+^ binding might be coupled to effects on the voltage-sensor equilibrium focus on models with one Ca^2+^ binding site per subunit. Although the 4 Ca^2+^ site formulation appears to provide an ability to discriminate between Model I and II, it seems likely that an 8 site formulation might not discriminate as well between the two categories of model. As such, the conclusions from the considerations in Figure 2 and 3 seem less strongly based than one would like.

5) There are various details in the results which are not mentioned. Why in some patches is there a non-zero offset during the gating current measurement? Was this dependent on Ca^2+^?

Finally, although the conclusions are not qualitatively novel and interaction between Ca binding and VSD activation has previously been incorporated in the HA model, the conclusion that the interaction is "strong" rather "weak" is an important distinction and its implications e.g. for the HA model are incompletely explored. For example, if Q_C_(V) is greatly shifted without a corresponding change in kinetics, then it suggests voltage-sensor activation cannot be described by a 2-state model. If the Q_C_(V) relation is strongly shifted by Ca but Q_O_-V is not then then VSD/Gate coupling (D) must be Ca-dependent (since D is defined by the difference between Q_C_ and Q_O_). Horrigan and Aldrich concluded that Q_O_ is not very Ca-dependent because the limiting slope of the Po-V relation (i.e. Q_A_-V) is only slightly shifted by Ca. We believe this is a result that has been essentially replicated by other labs. So trying to reconcile a large Ca-dependent shift in Q_C_(V) with previous data about the Ca-dependence of Po would likely require fundamental changes in mechanistic assumptions that are not addressed.

Regarding the assessment of the implications of the present results in comparison to Horrigan and Aldrich, it is not sufficient to simply claim the issue is resolved since the present results confirm earlier results from this same lab. That is ignoring the issue. Some assessment of whether there might be technical or analytical differences in each study would be minimally required. Taking both sets of results at face value, it would be helpful to provide readers with an objective comparison of the similarities and differences in what each group was measuring and how the gating current was determined. Although it may seem an unlikely explanation, there is also species difference and a difference in expression system. Although *hSlo1* and *mSlo1* are almost identical, there are unusual cases where single residue differences account for profound species differences. That can probably be discounted here, but it wouldn't hurt to mention it. Overall, the idea that there is some interaction between the cytosolic domain and the voltage sensor is not a new idea, but the question is the extent of the coupling. It should be kept in mind that, although the *Aplysia* structures have focused attention on this topic, the liganded *aSlo1* structure included both high Mg and high Ca, and it is not unexpected that in the presence of Mg some interaction between the CTD and VSD would be observed. But structural information has not yet revealed that Ca alone may stabilize linkages between the cytosolic domain and the voltage-sensor.

Overall, it is anticipated that the issues above can largely be addressed without new experiments, and it is hoped that some of the requested analysis may already be available. The reviewers feel that the observations in this paper are of merit, although currently difficult to interpret mechanistically. It is hoped that the requested changes will place this work on a stronger foundation that will help guide future work.

Reviewer #1:

This study examines the functional interaction between Calcium binding and voltage-sensor activation in BK channels, by measuring gating current at different [Ca^2+^] in the presence and absence of Ca^2+^ binding site mutations in RCK1 or RCK2 domains. The subject is timely given recent structural evidence that interactions between the cytoplasmic and voltage-sensor domains of Slo1 are modulated by Ca binding, and speculation that such interaction could make important contributions to Ca-dependent activation. Indeed, the authors conclude that direct interaction between Ca binding and voltage-sensor activation is strong, producing a ~26-fold increase in the equilibrium constant for voltage-sensor activation in saturating Ca^2+^. In addition, they find that Ca-binding to RCK1 and RCK2 make comparable and independent contributions to voltage sensor activation, and that the binding of Ca^2+^ to one subunit or site alters activation of all 4 voltage-sensors in a concerted manner. These conclusions have potentially important implication for understanding of BK channel gating mechanisms, with the caveat that some conclusions also appear to contradict previous studies making it more difficult to assess their significance.

The manuscript is generally well written, and the data are clearly described and appear to be of high quality. However, I think some additional analysis may be needed to help clarify differences between these and previous gating current results, and to better define the extent to which the data can discriminate between various models of interaction between Ca^2+^ binding sites and voltage sensors.

1) Charge movement equilibria and kinetics:

The authors find that saturating Ca produces a -143 mV Q_C_(V) shift, similar to a previous report from the same lab (Carrasquel-Ursulaez et al., 2015), but much larger than that reported by Horrigan and Aldrich (2002). Since there is no obvious explanation for these contradictory findings, the claim that "this result resolves a previous debate regarding the magnitude of the Ca^2+^-driven shift…" does not appear reasonable. While I am not certain what criteria could resolve such a debate, for the authors' merely to replicate their previous findings does not seem sufficient in the absence of a clear source of error. Perhaps the best that can be done at this stage is to provide a detailed description of the data and analysis so that the results of the two groups can be adequately compared. Therefore, I think it is important to see a more complete description of the gating current kinetics including their voltage-dependence. One reason is simply that Q_C_ is estimated by fitting the ON current with an exponential function. Thus, it is conceivable that the kinetic analysis could contribute to differences in Q_C_; and it would therefore be helpful to see examples of fits used to determine Q_C_ in the presence and absence of Ca^2+^. It also would be interesting to know whether a slow component of ON current and a slow increase in OFF charge with pulse duration (Qp) are observed in saturating Ca^2+^. Horrigan and Aldrich reported a large difference between Q_C_(V) and the steady state Q-V relationship (Q_SS_-V) in 70 μm Ca and prominent slow components of ON current and Qp (when measured at voltages near V_H_), consistent with a transition between Q_C_ and Q_SS_ when channels open. In the present study, however, if Q_C_(V) is strongly shifted by Ca^2+^ then one might expect less difference between Q_C_ and Q_SS_ and little or no slow components of ON current and Qp. Another expectation is that if Ca produces a large Q_C_(V) shift then it should also produce large changes in the time constant of fast charge movement (τ_fast_). Therefore, it would be helpful to compare τ_fast_(V) relations in the presence and absence of Ca. Horrigan and Aldrich observed a small Ca-dependent shift in the τ_fast_(V) curves, comparable to that of Q_C_(V), mainly due to an ~2-fold slowing of OFF charge movement. Surprisingly, the present study also describes small changes in kinetics including little effect of Ca on ON kinetics and a 2.5-fold slowing of the fast component of OFF kinetics in 10 μm Ca at -90 mV (Figure 2—figure supplement 1). How can these effects be reconciled with the propose 26-fold change in equilibrium constant? Do the shapes of the τ_fast_(V) curves provide any clues? Addressing these questions would help support the conclusions and gating models.

2) Gating Models:

Whether Ca binding to one subunit promotes activation of a single voltage sensor (Scheme I) or all 4 voltage-sensors through a concerted mechanism (Scheme II) is tested by comparing the ability of the corresponding gating schemes to fit Q_C_(V) curves in different Ca. Importantly Q_C_(V) curves shift with little change in shape. Therefore, Scheme I, which predicts a double-Boltzmann Q_C_(V) at intermediate Ca is qualitatively inconsistent with the data while Scheme II, which predicts a sum of 5 Boltzmanns, can better account for the shape of the curves. However, I don't think this is an adequate test of the two mechanisms. Both models incorporate a single Ca site per subunit, inconsistent with the conclusion from Figure 4 that both sites contribute equally to voltage sensor activation. An extended version of scheme I with 2 binding sites would predict Q_C_(V) curves that are essentially a sum of 3 Boltzmann functions and should therefore better fit the data. The question that should be addressed is whether the data can discriminate between versions of Schemes I and II with two binding sites per subunit.

A related issue that should be clarified is the criteria used to discriminate between fits. The authors indicate that "The best model fitting is that achieving the lowest AIC values". However, they conclude that the extended version of Scheme II "does not produce better fits to 𝑄C(𝑉, [Ca^2+^]) according to the AIC criteria" – even though the extended version appears to produce lower AIC values (Table 2) than the original version (Table 1). Presumably this is considered an insignificant difference. But there is no explanation of how to determine what is a significant difference.

3) Modeling assumptions:

As I understand it, the equations (4 and 6) describing the Q_C_(V) relations for Schemes I and II are based on the assumption that channels equilibrate between Ca-bound states while the channel is closed and voltage sensors are not activated – and that this distribution is not altered when gating current is measured because Ca binding is slower than voltage-sensor activation. While I agree that this is probably a reasonable assumption, I have some concern that it might not be. The authors argue "the Ca^2+^-binding rate constant estimated for BK channel is about 108 M^-1^s^-1^(Hou et al., 2016) implying that at 10 μM internal Ca^2+^ the association time constant is 1 ms. Thus, Ca^2+^ binding at this Ca^2+^ concentration proceeds at a pace about 33-fold slower than the voltage sensor movement (~30 μs)." I think this is an oversimplification and not accurate. First, the time-constant for Ca equilibration is also going to depend on the dissociation rate (~1560 s^-1^ for the low affinity site). Second, in the case of Scheme II, where subunits cannot be treated independently, the forward rate constant between Ca bound states is not simply equal to the association rate for a single binding site (1000 s^-1^) – e.g. it would be 4000 s^-1^ for the transition from zero to one Ca bound. So taken together, I think Ca-equilibration may only be ~4-5 slower than voltage-sensor movement. Is that enough of a difference?

Reviewer #2:

This manuscript shows a tour de force measurement of voltage dependence of gating charge movement (Q-V) during BK channel activation at various Ca concentrations ([Ca]). The large shift of the Q-V relation (-150 mV) in response to [Ca] increase from 0 to saturation (100 µM) suggests that Ca binding to the cytosolic gating ring alters voltage sensor activation. The authors conclude that the binding of Ca in each of the four subunits alters voltage sensor activation in all four subunits by fitting Q-V relations to different models. Mutational ablation of either of the two high affinity Ca^2+^ binding sites in each subunit reduces Ca^2+^ dependent shift of the Q-V relation by ~50%, suggesting that Ca^2+^ binding to either site contributes an equal amount of free energy in affecting voltage sensor activation. The authors made some conclusions based on these striking data, which are discussed as follows.

1) Results section: "In this manner, we determine only the gating charge displaced before the BK channel opening." This statement could be supported by showing ionic currents along with gating currents at the same voltages. The comparison in high [Ca] (≥10 µM) would be particularly helpful.

2) Figure 1 seems to show that as [Ca] increases the size of gating currents becomes smaller. Does this suggest that the total amount of fast charge movement diminishes at high [Ca]? What is the reason for this phenomenon and how is this related to the Ca-dependent shift of Q-V?

3) The authors interpret their data based on the Horrigan and Aldrich (HA) allosteric model of voltage and Ca^2+^ dependent activation, despite a larger Q-V shift in response to [Ca] increase is observed than that by HA. As the authors point out, while this study suggests that Ca^2+^ binding to either of the RCK1 site or Ca Bowl affects VSD equally, previous studies showed that Ca^2+^ binding only to the RCK1 site affects voltage dependence of ionic currents (G-V). Do these differences in experimental observation suggest any novel mechanism of BK channel gating that requires a modification of the HA model? In fact, although the authors suggest that HA Model II in Figure 2 fits their data better than HA Model I, it seems that neither model fits the data well (Figure 3).

4) Ca binding in one subunit equally affects all subunits, which suggests an allosteric interaction between Ca^2+^ binding and voltage sensor activation. It is not clear if Ca binding in one subunit alters all 4 cytosolic domains such that the entire gating ring interacts with all 4 VSD, or Ca binding in one subunit alters one VSD, which in turn affects all three other VSD's. There is also no evidence to distinguish if such an allosteric mechanism for Ca^2+^ binding to affect VSD activation is mediated by the covalent pulling of the C-linker or non-covalent interaction between the top of CTD and the bottom of VSD. Although based on previously published cryo-EM structures the authors favor the mechanism of the non-covalent VSD-CTD interactions, the present data seems not sufficient to exclude the contribution by the other mechanism.

Reviewer #3:

This is an important paper that addresses the issue of how Ca^2+^ binding to the cytosolic gating ring structure of BK channels allosterically couples to the voltage-sensor domains of BK channels, thereby ultimately influencing channel activation. One of the earlier papers on this topic (Horrigan and Aldrich, 2002) presented evidence that Ca^2+^ binding had only weak effects on voltage sensor equilibrium. More recent work has suggest that this may not be the case, and structures of the *Aplysia* Slo1 channel have supported the idea that there may be important links between the cytosolic domain and the voltage-sensors. Here, the authors present compelling evidence that Ca-binding to the cytosolic domain does allosterically influence VSD equilibrium and, furthermore, that each of the two Ca binding sites on a single BK α subunit independently, and largely additively, influence the VSD equilibrium.

The authors undertake an analysis of a potential mechanism explaining their data and reveal that occupancy of any binding site allosterically influences the VSD equilibrium of all subunits (their Scheme II), rather than only a single associated VSD (Scheme I). This requires that binding to any individual site exerts some concerted effect through all four VSDs.

Overall, the manuscript is clearly written and the data and analyses are solid. There are a few places that some clarifications in language are required. Also, a potential alternative conception of how 0-4 Ca^2+^ (actually, 0-8) may incrementally influence VSD equilibrium is mentioned, which the authors might want to consider.

[Editors' note: further revisions were requested prior to acceptance, as described below.]

Thank you for resubmitting your work entitled "Calcium-driven regulation of voltage-sensing domains in BK channels" for further consideration at *eLife*. Your revised article has been favorably evaluated by Richard Aldrich (Senior Editor), a Reviewing Editor, and three reviewers.

The manuscript has been improved but there are some remaining issues that need to be addressed before acceptance, as outlined below:

Details regarding the remaining concerns were initially developed in the comments from reviewer #1, but all reviewers were in agreement that, if these concerns are addressed, it will greatly improve the paper and provide a better framework for evaluating the implications of the present work without the context of earlier work. As a general comment, several of the arguments developed in the responses to the initial reviews would be useful in clarifying for readers what was done in this paper, and the rationale for various inferences (point 4 below). In particular, both figures that were included in the response to reviewers would be valuable additions as supplementary figures. Similarly, although the new manuscript includes the analysis from the two-high-affinity binding site model in Table 1, readers may find it odd that fits from these models are not included on main figures (or at least in figure supplements) in the paper. In particular, as reviewer #1 notes, although Table 1 supports the view that Model II does better than Model I, even for the two Ca site formulation, visually comparing these in fits to the Q-V and V_H_-Ca curves would give us a better feel for how well the data may discriminate between Model I and Model II (point 2 below). We therefore recommend including comparisons of fits on relevant panels, either in the main figures or, if that seems too cluttered, perhaps in supplementary figures.

There are four main categories of concern/recommendations corresponding to the itemization of reviewer #1.

1) Although the new information regarding gating current decay time constants is a valuable addition to the paper, there are aspects of this data that maintain the concern that Ca-equilibration with subunits with active voltage sensors may impact on the gating current decay time course, particularly at intermediate Ca. In particularly, the separation of calculated Ca equilibration rates and the observed time constants may not be as distinct as one might like. This might potentially impact the steepness of the V_H_-Ca relationship. I would recommend that the figure included in the rebuttal that plots the z∂ and residuals as a function of ratio of τ_Ca binding_/τ_gating currents_ be made into a supplementary figure, but with the addition that the analysis be done both using the first 30 μs and also the first 100 μs for comparison. Reviewer #1 also includes an additional suggestion for such a figure. Please also check whether the discontinuity in the plot of residuals observed at τ ratios of about 4 and 5 is a feature of the simulation or something else. Such a discontinuity seems a bit odd for a simulated data.

Also pertinent to this issue, two reviewers point out that the example gating current decay time constants from the new Figure 1A and 1B may not be fully consistent with the averaged data in Figure 1C. E.g., the time constant in 1B is above 0.8 of that in 1A, while at +160 mV in 1C, the time constants at 100 μM are about 0.5 that at 0 μM.

2) Showing the fits to the two Ca binding site formulation needs to be included, as mentioned above. Even if Model I and Model II give fits that extensively overlap, it is useful to know this. Similarly, a comparison of how much different the fits are with single site and two site Ca binding formulations would be helpful.

3) There appear to be some discrepancies in the properties of OFF gating current time constants that need to be addressed.

4) As stated at the outset, there is information in the response to reviewers that should be included in the manuscript to help guide readers.

Overall, all of these issues can presumably be addressed without new experiments. We hope the authors will agree that these recommendations will strengthen an interesting paper and better help the intended audience place this work within the context of previous work on BK channel gating.

Reviewer #1:

The authors' response and modifications to the manuscript are helpful and address many of my questions. However, there are still some issues that require clarification.

1) Appendix, subsection “Assumptions and model predictions”, third paragraph: "Ca^2+^ binding at this Ca^2+^ concentration proceeds at a pace about 33-fold slower than the voltage sensor movement (~30 μs)."

The new kinetic data (Figure 1C) indicates that the gating current time constant is actually as slow as 75 μs near V_H_ (i.e. slower than 0 Ca). That, and the simulated effect of Ca-binding kinetics in response to my previous review, have increased concerns about the assumption that Ca binding does not re-equilibrate during gating current measurements at intermediate [Ca], and I think this issue requires some additional analysis. This assumption is critical for modeling the Ca-dependence of the Q-V data and distinguishing between Schemes I and II. The simulation indicates that if Ca binding can re-equilibrate, then Q-Vs predicted by Scheme I will become more Boltzmann-like, potentially making it more difficult to distinguish Schemes I and II. In addition, I wonder if re-equilibration could contribute to the remarkable steepness of the V_H_-Ca relation between 5 and 10 μm Ca – a feature that is not well described by any of the models. That is, re-equilibration should increase the V_H_ shift (since voltage-sensor activation will enhance Ca-binding) and if Ca re-equilibration is faster at higher [Ca] then it would tend to increase the steepness of the curve

The 75 μs gating current time constant in Figure 1C increases my concern because it is potentially closer to the time constant of Ca binding. For example, at the [Ca] used in the simulation (15 μM), the on-rate at a single binding site would be 1500 s^-1^, or 6000 s^-1^ for the transition from zero to one Ca bound with 4 binding sites, implying a time constant less than 166 μs (depending on off-rate). Thus, it seems possible that Ca-binding could be only ~2 fold slower than voltage sensor activation under some conditions. The simulation seems to suggest this is probably not a problem since there is little change in the Q-V fit parameters when the Ca-binding time constant is 3-fold or more slower than gating current. However, I have two concerns about the simulation analysis: (1) the Q-Vs were measured by fitting the first 30 μs of the simulated gating current, whereas the experimental data were measured from the first 100 μs. This could lead the simulation to underestimate the potential contribution of Ca re-equilibration to the data. (2) It doesn't seem reasonable to fit the simulated Q-Vs with a single Boltzmann function if the assumption to be tested (no re-equilibration) predicts a double-Boltzmann Q-V. In other words, although the simulation shows little change in the Q-V fit parameters when the Ca-binding time constant is 3-fold or more slower than gating current, I wonder if this indicates that there is no change in the Q-V over this range, or merely that the fits to a single Boltzmann function are all equally bad. I think the simulation analysis should be repeated by fitting the initial 100 μs of the gating current, and fitting the resulting Q-Vs to the prediction when there is no re-equilibration of Ca (i.e. double Boltzmann). Presumably, such fits, or showing the actual Q-Vs, would better define the conditions under which the model assumptions are valid. Plotting V_H_ might also be useful.

2) Although I accept the conclusion, based on Table 1, that Scheme II fits the data better than Scheme I even with 2 binding sites per subunit, I think it would be good to show the fits for the 2-site models, at least in the supplement. One reason, as I suggested in my initial review, is that I don't think it is reasonable to dismiss Scheme I based on the qualitative difference in Q-V shape for a one-site model, when such a model is model is inconsistent with the data in Figure 5. That is simply a "straw man" argument. But, mainly, as noted by another reviewer, none of the models fit very well, suggesting neither Schemes are the final answer. In such a situation, I think it is important to show the fits for future reference.

3) The authors partially addressed my question about gating current kinetics with Figure 1C. The time constants for the ON currents do appear to shift along the voltage-axis in response to Ca, consistent with the Q-V. But they didn't address my question regarding the relatively small 2.5-fold Ca-dependent change in OFF kinetics in Figure 2—figure supplement 1. The authors suggest, based on Figure 1C that the main effect Ca is to slow OFF kinetics, but Figure 2—figure supplement 1 isn't consistent with that prediction. Presumably if ON time constants are decreased ~2-fold by Ca then OFF time constants should increase ~10 fold to account for the large E-factor. Are the data representative? Can OFF kinetics included in the time constant plot?

4) Finally, many of the responses to reviewers are fine, but should be incorporated in the manuscript. Requests for clarification generally refer to issues that need to be better explained not only to the reviewers but also to the intended audience of the paper. For example, when I requested clarification in the Materials and methods section regarding the grounding scheme, I meant that such details should be included in the Materials and methods section. Similarly, the author's clarification that representative traces in Figure 2A are from different patches with the exception of 0 and 100 Ca is helpful, but should be described in the figure legend.

Reviewer #2:

The revised manuscript is improved, and all my questions are addressed.

Reviewer #3:

The authors have responded in generally satisfactory fashion to many of the major issues raised by the reviewer. The includes the kinetics of Q_C_activation, how Q_C_was measured, the apparent Ca^2+^-dependence of gating current amplitude, the impact of including 2 types of Ca^2+^ binding sites in the model fitting, and a more balanced consideration of how the present results fit with previous work.

The manuscript and new changes still suffer from abundant English usage errors, including some where the meaning of the phrasing is not clear.

The new Figure 1 nicely addresses the issue of what is actually being measured and whether the Q_C_measurements might include contamination from later voltage sensor movements. Furthermore, the figure address the topic of the Ca^2+^ dependent of Q_C_kinetics. There is one issue that persists in this data. The time constant in 1A (0 Ca^2+^) is 52 μs, while in 1B (100 μM) it is 42 μs, both at +160 mV. Examination of the averaged data in panel C show that at +160 mV, μI_G-ON_ is a bit larger than 0.06 ms for 0 Ca^2+^, generally consistent with the trace in 1A, while for 100 μM it appears to be a little above 0.025 ms. The values for the displayed example at 100 μM seems to be well outside the error bars, than one might expect. Perhaps something needs to be checked here.

[Editors' note: further revisions were requested prior to acceptance, as described below

Thank you for resubmitting your work entitled "Calcium-driven regulation of voltage-sensing domains in BK channels" for further consideration at *eLife*. Your revised article has been favorably evaluated by Richard Aldrich (Senior Editor), a Reviewing Editor, and three reviewers.

The manuscript has been improved but there a few remaining issues that need to be addressed before acceptance, as outlined below:

Please address reviewer #1's concerns about the off gating charge and correct the minor language issues. I believe that these will be easy for you to address.

Reviewer #1:

The revised manuscript is improved and adequately addresses most of my concerns. However, the authors expressed some confusion over my questions regarding the apparent discrepancy in OFF gating current kinetics, which I will attempt to clarify.

"We are somewhat confused about the reviewer 1 comment 3. As stated in Results the 𝜏𝐺−𝑂𝑁(𝑉) curves were fitted to a two-state model (𝜏(𝑉)=1/(𝛼(𝑉)+𝛽(𝑉)) where the forward (𝛼) and backward (𝛽) rate constants represent the resting-active (R-A) transitions of the voltage sensors determining the equilibrium constant of the VSD activation (𝐽(𝑉)=𝛼(𝑉)𝛽(𝑉)). Our statement that the main effect of Ca^2+^ is to slow down the OFF kinetics is based on the determination of 𝛼(𝑉) and 𝛽(𝑉) using the values of 𝜏𝐼𝐺−𝑂𝑁(𝑉) and J(V). This calculation indicates that 𝛽(𝑉) increase 10.9-fold when the Ca^2+^ concentration is increased to 100 μM (see legend in Figure 1), in a reasonable agreement with the large allosteric factor E. It is unclear to us why the reviewer is referring to Figure 2—figure supplement 1, a figure that is not directly comparable to Figure 1C."

My questions about gating current kinetics (Comments #3 in previous review, #1 in original review) relate in large part to the data in Figure 2—figure supplement 1. I asked whether the difference in gating current kinetics shown for 0 and 10 μm Ca in Figure 2—figure supplement 1 are consistent with a large Q-V shift (as observed in 10 Ca). In particular, I noted that there is a relatively small change from 0 to 10 Ca in the time constant of the fast component of OFF current at -90 mV, which doesn't seem compatible with a large Q-V shift. The fast OFF component at -90 mV should be a relatively direct measure of the backward rate β (i.e. τ~=1/β). If there is a small change in ON kinetics and large Q-V shift, then there should be a large change in β and hence OFF kinetics. The authors make essentially the same prediction based on their fits to the ON kinetics in Figure 1C. But they do not include OFF data to directly confirm the prediction or address my concern.

"We would like to include the OFF time constant plot requested by the reviewer, but these kind of experiments are in extremely difficult to do at high Ca^2+^,.…"

I do not think that measurements of OFF kinetics at multiple voltages are necessary. Measurements at a single voltage like -90 mV should suffice. If OFF kinetics have not or cannot be measured at 100 Ca, then I suggest adding the ON kinetics at 10 Ca to Figure 1C together with OFF data at -90 mV for 0 and 10 Ca. Based on Figure 2—figure supplement 1, such data already exists.

Reviewer #2:

The authors took the suggestions of the last review and the revised manuscript includes more complete data and balanced description. I have no further comments.

Reviewer #3:

Overall, the authors have addressed most or all of the issues that have been raised in previous reviews. It is still unclear why the results here differ from those of Horrigan and Aldrich, but the present paper now thoroughly documents the results, and provides more detailed consideration of the assumptions and analysis that lead the authors to their conclusions. Furthermore, the paper now addresses the unresolved issues in a generally more scholarly fashion. Although questions remain, this paper provides a valuable contribution to the issue of CTD-VSD coupling in BK channels.

---

## [Author Response]

Reviewer #1:[…]1) Charge movement equilibria and kinetics:The authors find that saturating Ca produces a -143 mV Q_C_(V) shift, similar to a previous report from the same lab (Carrasquel-Ursulaez et al., 2015), but much larger than that reported by Horrigan and Aldrich, 2002. Since there is no obvious explanation for these contradictory findings, the claim that "this result resolves a previous debate regarding the magnitude of the Ca^2+^-driven shift…" does not appear reasonable. While I am not certain what criteria could resolve such a debate, for the authors' merely to replicate their previous findings does not seem sufficient in the absence of a clear source of error. Perhaps the best that can be done at this stage is to provide a detailed description of the data and analysis so that the results of the two groups can be adequately compared. Therefore, I think it is important to see a more complete description of the gating current kinetics including their voltage-dependence. One reason is simply that Q_C_ is estimated by fitting the ON current with an exponential function. Thus, it is conceivable that the kinetic analysis could contribute to differences in Q_C_; and it would therefore be helpful to see examples of fits used to determine Q_C_ in the presence and absence of Ca^2+^.

During the last Biophysical Meeting, one of us (RL) has the opportunity to discuss at length with Frank Horrigan the possible reasons regarding the discrepancies between the HA 2002 results and ours. There are two differences between the experimental procedures followed by HA 2002 and ours. First, HA used the mouse BK clone, and we worked with the human BK; and second, the expression systems are different. HEK transfected cells in the case of HA and RNA injection into *Xenopus oocytes* in our case. Since the mBK share a high degree of identity (96%), it is difficult to reconcile the differences in Ca^2+^ binding and voltage sensor coupling based in the fact that we used to different BK clones. The possibility that the strength of coupling is different in different expression systems (e.g., differential modulation) is a possibility, however, outside the scope of the present communication. Taking the advice of the Senior Editor, we have added a paragraph in the Discussion indicating the possible sources of the differences between the HA 2002 results and the results presented here.

We are now stressing the point that although we cannot give a plausible explanation to these differences, the data presented in this paper were analyzed in deep and showed no hints of possible artifacts that can obscure their interpretation.

Below, we would like to convince reviewer #1 by giving a detailed analysis of our results that indeed we are estimating only the gating charge displaced between closed states (Q_C_).

As requested by reviewer #1, in Figure 1A-B we now show that both in the absence and in the presence of 100 µM Ca^2+^ the macroscopic K^+^ currents activate with a delay that is approximately 160 µs and 84 µs, respectively. In our experiments, we have isolated the gating charge corresponding to voltage sensor activation while BK channels are closed (Q_C_) by fitting the first 100 µs with an exponential function. We argue that since during that period the channels are closed regardless of the internal Ca^2+^ concentration, we are determining the direct interaction between Ca^2+^ binding and the voltage sensor.

It also would be interesting to know whether a slow component of ON current and a slow increase in OFF charge with pulse duration (Qp) are observed in saturating Ca^2+^. Horrigan and Aldrich reported a large difference between Q_C_(V) and the steady state Q-V relationship (Q_SS_-V) in 70 μm Ca and prominent slow components of ON current and Qp (when measured at voltages near V_H_), consistent with a transition between Q_C_ and Q_SS_ when channels open. In the present study, however, if Q_C_(V) is strongly shifted by Ca^2+^ then one might expect less difference between Q_C_ and Q_SS_ and little or no slow components of ON current and Qp.

We agree with the reviewer, and our data fulfil their prediction. As can be appreciated in the new Figure 1B there is no appreciable slow component in I_G-ON_ at high internal Ca^2+^ concentration. However, Figure 1B shows that after one ms with the voltage held at 160 mV about 80% of the BK channels are open and as shown in Figure 2A the OFF gating current recorded at -90mV are slowed considerably in agreement with the prediction of the allosteric model in which the OFF response should exhibit a time course similar to that of the time constant of the channel deactivation process. In the Figure 2—figure supplement 2 we compared Q_C_ and Q_OFF_ at low and high internal Ca^2+^. Since the OFF gating charge cannot be well estimated from the OFF gating current recorded at -90mV, we obtained two experiments at 100 µM internal Ca^2+^ concentration and recorded the OFF gating current at -150mV and -200mV. As expected from the large shift to the left along the voltage axis of Q_C_ at saturating internal Ca^2+^ concentrations, there is no difference between these two curves measuring the OFF gating current 2 ms after the onset of the voltage pulse.

Another expectation is that if Ca produces a large Q_C_(V) shift then it should also produce large changes in the time constant of fast charge movement (τ_fast_). Therefore, it would be helpful to compare τ_fast_(V) relations in the presence and absence of Ca. Horrigan and Aldrich observed a small Ca-dependent shift in the τ_fast_(V) curves, comparable to that of Q_C_(V), mainly due to an ~2-fold slowing of OFF charge movement. Surprisingly, the present study also describes small changes in kinetics including little effect of Ca on ON kinetics and a 2.5-fold slowing of the fast component of OFF kinetics in 10 μm Ca at -90 mV (Figure 2—figure supplement 1). How can these effects be reconciled with the propose 26-fold change in equilibrium constant? Do the shapes of the τ_fast_(V) curves provide any clues? Addressing these questions would help support the conclusions and gating models.

The fast time constants together with their respective Q_C_(V) curves are now plotted in Figure 1C. Note that the τ_fast_(V) curve at high Ca^2+^ concentration is leftward shifted and it maximum coincides with the half voltage of the Q(V) as expected from a two-state model describing the resting-active equilibrium of the voltage sensor.

2) Gating Models:Whether Ca binding to one subunit promotes activation of a single voltage sensor (Scheme I) or all 4 voltage-sensors through a concerted mechanism (Scheme II) is tested by comparing the ability of the corresponding gating schemes to fit Q_C_(V) curves in different Ca. Importantly Q_C_(V) curves shift with little change in shape. Therefore, Scheme I, which predicts a double-Boltzmann Q_C_(V) at intermediate Ca is qualitatively inconsistent with the data while Scheme II, which predicts a sum of 5 Boltzmanns, can better account for the shape of the curves. However, I don't think this is an adequate test of the two mechanisms. Both models incorporate a single Ca site per subunit, inconsistent with the conclusion from Figure 4 that both sites contribute equally to voltage sensor activation. An extended version of scheme I with 2 binding sites would predict Q_C_(V) curves that are essentially a sum of 3 Boltzmann functions and should therefore better fit the data. The question that should be addressed is whether the data can discriminate between versions of Schemes I and II with two binding sites per subunit.

To answer this query, we have now included both types of models with one Ca^2+^ per subunit and two Ca^2+^ per subunit in Table 1. As the reviewer can appreciate the likelihood of Model II (one Ca^2+^ affects all voltage sensors) is always orders of magnitude larger (see values of ℒ_i_ and w_i_) than that of the Model I. We conclude that the data can discriminate between versions of Models I and II with two binding sites per subunit.

*A related issue that should be clarified is the criteria used to discriminate between fits. The authors indicate that "The best model fitting is that achieving the lowest AIC values". However, they conclude that the extended version of Scheme II "does not produce better fits to* 𝑄*C(*𝑉*, [Ca2+]) according to the AIC criteria" – even though the extended version appears to produce lower AIC values (Table 2) than the original version (Table 1). Presumably this is considered an insignificant difference. But there is no explanation of how to determine what is a significant difference.*

The data of the new Table 1 shows that the Model II with two Ca^2+^-sites per subunit with or without cooperativity is slightly better than Model II with one Ca^2+^-site per subunit. However, we think that this difference is nor statistically significant.

3) Modeling assumptions:As I understand it, the equations (4 and 6) describing the Q_C_(V) relations for Schemes I and II are based on the assumption that channels equilibrate between Ca-bound states while the channel is closed and voltage sensors are not activated – and that this distribution is not altered when gating current is measured because Ca binding is slower than voltage-sensor activation. While I agree that this is probably a reasonable assumption, I have some concern that it might not be. The authors argue "the Ca^2+^-binding rate constant estimated for BK channel is about 108 M^-1^s^-1^ (Hou et al., 2016) implying that at 10 μM internal Ca^2+^ the association time constant is 1 ms. Thus, Ca^2+^ binding at this Ca^2+^ concentration proceeds at a pace about 33-fold slower than the voltage sensor movement (~30 μs)." I think this is an oversimplification and not accurate. First, the time-constant for Ca equilibration is also going to depend on the dissociation rate (~1560 s^-1^ for the low affinity site). Second, in the case of Scheme II, where subunits cannot be treated independently, the forward rate constant between Ca bound states is not simply equal to the association rate for a single binding site (1000 s^-1^) – e.g. it would be 4000 s^-1^ for the transition from zero to one Ca bound. So taken together, I think Ca-equilibration may only be ~4-5 slower than voltage-sensor movement. Is that enough of a difference?

Yes, calcium binding time constant 4-5 times slower than the gating current time constant is slow enough to consider that calcium binding equilibrium does not change during the time the gating currents are measured (see Author response image 1). To answer this question, we performed simulations of the gating currents using the model that assumes that Ca^2+^-binding sites and voltage sensors can only interact within the same subunit. The results shown in Author response image 1 indicates that the calcium binding is slow enough to safely assume that the calcium equilibrium is not altered during the short time it takes to measure the gating currents. The figure shows that if we assume that Ca^2+^ only affects the voltage sensor of the same subunit, we would have expected a measurable change in the Q(V) curve slope (zδ) something that is not detected experimentally.

**Author response image 1. respfig1:** Boltzmann sigmoidal curve fits of the Q(**V**) curves simulated for a BK model where the occupancy of the calcium binding site of one subunit alters the voltage sensor of only the same subunit. Using the on binding rate constant of Hou et al., 2016 kb = 1.81⋅108 M^-1^s^-1^ and the unbinding rate constant ku = 650 s^-1^, calculated from the calcium dissociation constant of 3.6⋅10-6. We fitted the simulated the gating current at 15 μM Ca^2+^ to a single exponential using the first 30 μs of the gating current decay. Q(**V**) data obtained by integration of the area under the exponential function were fitted using a Boltzmann function. The sum of squared residual indicates that the fit is good for the calcium-binding time constant one hundred times shorter than gating current time constant. Also, we recover the zd value (0.58) we used for the simulation, only when the Ca^2+^ binding is 20 to 200 times faster than the voltage sensor kinetics. The high sum of squared residuals and the low zδ value obtained for calcium binding time constant three or more times slower than the gating currents time constant (pointed with arrows) indicates that the Q(**V**) curve is not a simple Boltzmann sigmoidal function.

Reviewer #2:This manuscript shows a tour de force measurement of voltage dependence of gating charge movement (Q-V) during BK channel activation at various Ca concentrations ([Ca]). The large shift of the Q-V relation (-150 mV) in response to [Ca] increase from 0 to saturation (100 µM) suggests that Ca binding to the cytosolic gating ring alters voltage sensor activation. The authors conclude that the binding of Ca in each of the four subunits alters voltage sensor activation in all four subunits by fitting Q-V relations to different models. Mutational ablation of either of the two high affinity Ca^2+^ binding sites in each subunit reduces Ca^2+^ dependent shift of the Q-V relation by ~50%, suggesting that Ca^2+^ binding to either site contributes an equal amount of free energy in affecting voltage sensor activation. The authors made some conclusions based on these striking data, which are discussed as follows.1) Results section: "In this manner, we determine only the gating charge displaced before the BK channel opening." This statement could be supported by showing ionic currents along with gating currents at the same voltages. The comparison in high [Ca] (≥10 µM) would be particularly helpful.

The data requested by the reviewer is now included in Figure 1A-B. In it, gating and macroscopic current records are compared at low and high internal (100 μM) Ca^2+^ concentration. As the reviewer can appreciate the gating current time course develops and is almost complete in both cases in the time interval comprised by the macroscopic current delay. Please see also the answer to the first comment of reviewer #1.

2) Figure 1 seems to show that as [Ca] increases the size of gating currents becomes smaller. Does this suggest that the total amount of fast charge movement diminishes at high [Ca]? What is the reason for this phenomenon and how is this related to the Ca-dependent shift of Q-V?

We understand the confusion of the reviewer since the experiments in Figure 1 (now is Figure 2) were badly described and the current gating records shown were not the most representative of the average at a given Ca^2+^ concentration. Actually, the records shown at the different Ca^2+^ concentrations are not from a single experiment. However, at all the Ca^2+^ concentrations shown in Figure 2, the control was always 0 Ca^2+^. So to answer the reviewer’s query, we did not find any decrease in gating current when going form 0 Ca^2+^ to the different Ca^2+^ concentrations tested. Accordingly, we have now included a new Figure 2 where the records obtained at 0 and 100 μM Ca^2+^ are from the same experiment.

3) The authors interpret their data based on the Horrigan and Aldrich (HA) allosteric model of voltage and Ca^2+^ dependent activation, despite a larger Q-V shift in response to [Ca] increase is observed than that by HA. As the authors point out, while this study suggests that Ca^2+^ binding to either of the RCK1 site or Ca Bowl affects VSD equally, previous studies showed that Ca^2+^ binding only to the RCK1 site affects voltage dependence of ionic currents (G-V). Do these differences in experimental observation suggest any novel mechanism of BK channel gating that requires a modification of the HA model? In fact, although the authors suggest that HA Model II in Figure 2 fits their data better than HA Model I, it seems that neither model fits the data well (Figure 3).

A modification of the HA model is not required. The importance of the results we are presenting resides in that they allow to distinguish between the two alternative models proposed in the HA 2002 paper something not possible without the gating current data obtained at different Ca^2+^ concentrations. Regarding the goodness of the fitting, we argue here that Model II statistically fits the data much better taking into account that the AIC = 2p – 2ln(L) and the shape of the Q(V)s approach at all Ca^2+^ concentrations a simple Boltzmann function. Given that the number of parameters in both models is the same, the likelihood (L) of Model II is orders of magnitude larger than the likelihood of Model I. Please see new Table 1 and answer to comment 2 of reviewer #1.

4) Ca binding in one subunit equally affects all subunits, which suggests an allosteric interaction between Ca^2+^ binding and voltage sensor activation. It is not clear if Ca binding in one subunit alters all 4 cytosolic domains such that the entire gating ring interacts with all 4 VSD, or Ca binding in one subunit alters one VSD, which in turn affects all three other VSD's. There is also no evidence to distinguish if such an allosteric mechanism for Ca^2+^ binding to affect VSD activation is mediated by the covalent pulling of the C-linker or non-covalent interaction between the top of CTD and the bottom of VSD. Although based on previously published cryo-EM structures the authors favor the mechanism of the non-covalent VSD-CTD interactions, the present data seems not sufficient to exclude the contribution by the other mechanism.

We think that the gating ring can be considered as a single structure where the four C-terminal domain interacts through non-covalent interactions. So, it is more parsimonious to think that binding of one Ca^2+^ should increase the whole gating ring diameter affecting all four voltage sensors. The reviewer is right, however, in calling our attention about possible alternative mechanisms intervening in producing the gating current results we are reporting. It is not possible at present to say how much are contributing gating ring-VSD non-covalent interactions and conformational changes mediated by the pulling of the C-linker mediated by the gating ring to the leftward shift of the Q(V) curved induced by increasing internal Ca^2+^ concentration. However, we recall here the fact that in BK channels voltage sensors are not domain-swapped concerning the pore domain as in voltage-dependent K^+^ channels, but the VSDs are domain-swapped with RCK domains in the gating ring. Therefore, binding of Ca^2+^ makes the RCK1 N-lobe pull on the S6 helix from its subunit whereas the modification of the contact surface between the RCK1 N-lobe with the voltage sensor of an adjacent subunit induces and outward displacement of the voltage sensor (Hite et al., 2017). These structural arguments make us favour the non-covalent interaction between the CTD and the VSD as the source of the coupling between these two structures. The reviewer is right, however, in that we cannot say for certain how much of this coupling is mediated by the C-linker in the same subunit. We have rewritten this section treating the subject more gingerly and discussing possible mechanisms that can explain how Ca^2+^ binding to one subunit may affect the voltage sensors in all subunits.

Reviewer #3:This is an important paper that addresses the issue of how Ca^2+^ binding to the cytosolic gating ring structure of BK channels allosterically couples to the voltage-sensor domains of BK channels, thereby ultimately influencing channel activation. One of the earlier papers on this topic (Horrigan and Aldrich, 2002) presented evidence that Ca^2+^ binding had only weak effects on voltage sensor equilibrium. More recent work has suggest that this may not be the case, and structures of the Aplysia Slo1 channel have supported the idea that there may be important links between the cytosolic domain and the voltage-sensors. Here, the authors present compelling evidence that Ca-binding to the cytosolic domain does allosterically influence VSD equilibrium and, furthermore, that each of the two Ca binding sites on a single BK α subunit independently, and largely additively, influence the VSD equilibrium.The authors undertake an analysis of a potential mechanism explaining their data and reveal that occupancy of any binding site allosterically influences the VSD equilibrium of all subunits (their Scheme II), rather than only a single associated VSD (Scheme I). This requires that binding to any individual site exerts some concerted effect through all four VSDs.Overall, the manuscript is clearly written and the data and analyses are solid. There are a few places that some clarifications in language are required. Also, a potential alternative conception of how 0-4 Ca^2+^ (actually, 0-8) may incrementally influence VSD equilibrium is mentioned, which the authors might want to consider.

[Editors' note: further revisions were requested prior to acceptance, as described below.]

Reviewer #1:The authors' response and modifications to the manuscript are helpful and address many of my questions. However, there are still some issues that require clarification.1) "Ca^2+^ binding at this Ca^2+^ concentration proceeds at a pace about 33-fold slower than the voltage sensor movement (~30 μs)."The new kinetic data (Figure 1C) indicates that the gating current time constant is actually as slow as 75 μs near VH (i.e. slower than 0 Ca). That, and the simulated effect of Ca-binding kinetics in response to my previous review, have increased concerns about the assumption that Ca binding does not re-equilibrate during gating current measurements at intermediate [Ca], and I think this issue requires some additional analysis. This assumption is critical for modeling the Ca-dependence of the Q-V data and distinguishing between Schemes I and II. The simulation indicates that if Ca binding can re-equilibrate, then Q-Vs predicted by Scheme I will become more Boltzmann-like, potentially making it more difficult to distinguish Schemes I and II. In addition, I wonder if re-equilibration could contribute to the remarkable steepness of the VH-Ca relation between 5 and 10 μm Ca – a feature that is not well described by any of the models. That is, re-equilibration should increase the VH shift (since voltage-sensor activation will enhance Ca-binding) and if Ca re-equilibration is faster at higher [Ca] then it would tend to increase the steepness of the curveThe 75 μs gating current time constant in Figure 1C increases my concern because it is potentially closer to the time constant of Ca binding. For example, at the [Ca] used in the simulation (15 μM), the on-rate at a single binding site would be 1500 s^-1^, or 6000 s^-1^ for the transition from zero to one Ca bound with 4 binding sites, implying a time constant less than 166 μs (depending on off-rate). Thus, it seems possible that Ca-binding could be only ~2 fold slower than voltage sensor activation under some conditions. The simulation seems to suggest this is probably not a problem since there is little change in the Q-V fit parameters when the Ca-binding time constant is 3-fold or more slower than gating current. However, I have two concerns about the simulation analysis: (1) the Q-Vs were measured by fitting the first 30 μs of the simulated gating current, whereas the experimental data were measured from the first 100 μs. This could lead the simulation to underestimate the potential contribution of Ca re-equilibration to the data. (2) It doesn't seem reasonable to fit the simulated Q-Vs with a single Boltzmann function if the assumption to be tested (no re-equilibration) predicts a double-Boltzmann Q-V. In other words, although the simulation shows little change in the Q-V fit parameters when the Ca-binding time constant is 3-fold or more slower than gating current, I wonder if this indicates that there is no change in the Q-V over this range, or merely that the fits to a single Boltzmann function are all equally bad. I think the simulation analysis should be repeated by fitting the initial 100 μs of the gating current, and fitting the resulting Q-Vs to the prediction when there is no re-equilibration of Ca (i.e. double Boltzmann). Presumably, such fits, or showing the actual Q-Vs, would better define the conditions under which the model assumptions are valid. Plotting VH might also be useful.

As suggested by reviewer #1 we have redone the simulation analysis considering the first 100 μ s of the gating current and 10 μM Ca^2+^ (please see new Figure 3—figure supplement 2). We have confirmed our previous conclusion that Ca^2+^ binding time constants 5-fold slower than the gating time constant suffice to consider that there is not Ca^2+^ re-equilibration during the time it takes to measure the gating currents. Figure 3—figure supplement 2 and the text are now included in the Appendix.

We performed a single Boltzmann curve fit in our simulations: Model I predicts that, at calcium concentrations close to the dissociation constant, the Q/V curves should have a minimum in the z_Q_ parameter and be poorly described with a sigmoidal Boltzmann function. However, as is observed in Figure 4, the Q/V curves are well described with a sigmoidal Boltzmann function and that the parameter z_J_ is the same for all calcium concentrations. It can be argued that this result is compatible with the Model I if it is accepted that the equilibrium kinetics of calcium is rapid. To estimate how fast this equilibrium has to be to produce well-described Q/V curves with a Boltzmann with invariant z_Q_, we performed the simulations shown in Figure 3—figure supplement 2. We found that to recover the experimental z_Q,_ the calcium-binding rate constant needs to be 100 times faster than that reported by Hou et al., 2016 exceeding the diffusion limit constraint. We conclude that the good fit to a single Boltzmann sigmoidal function to our experimental Q/V curves is not due to fast calcium re-equilibration but due to the failure of Model I to describe our data. This conclusion is the same for analysis based on the first 30 μs or the first 100 μs of the simulated gating currents. V_H_ results are not included in the figure since they would not contribute to discriminate between the alternative interpretations of our findings.

2) Although I accept the conclusion, based on Table 1, that Scheme II fits the data better than Scheme I even with 2 binding sites per subunit, I think it would be good to show the fits for the 2-site models, at least in the supplement. One reason, as I suggested in my initial review, is that I don't think it is reasonable to dismiss Scheme I based on the qualitative difference in Q-V shape for a one-site model, when such a model is model is inconsistent with the data in Figure 5. That is simply a "straw man" argument. But, mainly, as noted by another reviewer, none of the models fit very well, suggesting neither Schemes are the final answer. In such a situation, I think it is important to show the fits for future reference.

As requested by the reviewer, Figure 4—figure supplement 1 now shows the fits of the data using the 2-site models, and parameters are provided in the new Table 1. Figure 4—figure supplement 1 is described and discussed in the Results (subsection “High-affinity Ca^2+^-binding sites in RCK1 and RCK2 domains contribute equally to the allosteric coupling between Ca^2+^ and voltage sensors”) and the Discussion section, respectively, of the new version of the manuscript. It is evident from the new fittings that due to the increases in the numbers of parameters – when using the two-site models, the differences between models are attenuated compared to the fitting to the data using a single Ca^2+^ site. However, Table 1 shows that the AIC values greatly favor Model II-two Ca^2+^ sites. The goodness of the different models can be better appreciated by plotting of z_J_ vs. Ca^2+^ (Figure 4D). Note that in Model I-two Ca^2+^ sites, reflecting the proportion of unliganded and Ca^2+^-saturated channels, z_J_ is Ca^2+^ dependent, reaching a minimum when the Ca^2+^ concentration is ~ 10 μM (Figure 4D and Figure 4—figure supplement 1C-D), whereas as found experimentally, z_J_ in Model II-Two Ca^2+^ sites z_J_ is independent of the Ca^2+^ concentration (Figure 4D).

We should note here that in revising all the Q(V)s at the different Ca^2+^ concentration, we have found that some of the data for 5 and 10 μM Ca^2+^ were not adequately analyzed and some of the Q(V)s were statistical outliers. Taking into account the five and four remaining Q(V) curves at 5 and 10 μM Ca^2+^, we found that the V_H_ undergoes a slight rightward shift from 115.7 ± 6 to 121.9 ± 3.8 mV and z_Q_ from 0.67 ± 0.04 to 0.63 ± 0.01 for 5 μM Ca^2+^; whilst for 10 μM Ca^2+^ V_H_ is shifted from 46.5 ± 14 to 67.3 ± 17 mV and z_Q_ from 0.71 ± 0.07 to 0.64 ± 0.08. The main consequence of this new analysis is that now the Model II fits the data better according to the AIC criteria value (see Table 1).

3) The authors partially addressed my question about gating current kinetics with Figure 1C. The time constants for the ON currents do appear to shift along the voltage-axis in response to Ca, consistent with the Q-V. But they didn't address my question regarding the relatively small 2.5-fold Ca-dependent change in OFF kinetics in Figure 2—figure supplement 1. The authors suggest, based on Figure 1C that the main effect Ca is to slow OFF kinetics, but Figure 2—figure supplement 1 isn't consistent with that prediction. Presumably if ON time constants are decreased ~2-fold by Ca then OFF time constants should increase ~10 fold to account for the large E-factor. Are the data representative? Can OFF kinetics included in the time constant plot?

We are somewhat confused about the reviewer 1 comment 3. As stated in Results the τIG-ONV curves were fitted to a two-state model (τ(V)=1/(α(V)+β(V)) where the forward (α) and backward (β) rate constants represent the resting-active (R-A) transitions of the voltage sensors determining the equilibrium constant of the VSD activation (JV = αVβV). Our statement that the main effect of Ca^2+^ is to slow down the OFF kinetics is based on the determination of αV and β(V) using the values of τIG-ONV and J(V). This calculation indicates that β(V) increase 10.9-fold when the Ca^2+^ concentration is increased to 100 μM (see legend in Figure 1), in a reasonable agreement with the large allosteric factor E. It is unclear to us why the reviewer is referring to Figure 2—figure supplement 1, a figure that is not directly comparable to Figure 1C.

We would like to include the OFF time constant plot requested by the reviewer, but these kind of experiments are in extremely difficult to do at high Ca^2+,^ and we cannot anticipate the time we will spend in them.

4) Finally, many of the responses to reviewers are fine, but should be incorporated in the manuscript. Requests for clarification generally refer to issues that need to be better explained not only to the reviewers but also to the intended audience of the paper. For example, when I requested clarification in the Materials and methods section regarding the grounding scheme, I meant that such details should be included in the Materials and methods section. Similarly, the author's clarification that representative traces in Figure 2A are from different patches with the exception of 0 and 100 Ca is helpful, but should be described in the figure legend.

The grounding scheme is now in Materials and methods and the clarification requested by the reviewer regarding the traces in Figure 2A is now included in the legend of the figure.

Reviewer #3:The authors have responded in generally satisfactory fashion to many of the major issues raised by the reviewer. The includes the kinetics of Q_C_activation, how Q_C_was measured, the apparent Ca^2+^-dependence of gating current amplitude, the impact of including 2 types of Ca^2+^ binding sites in the model fitting, and a more balanced consideration of how the present results fit with previous work.The version of the manuscript which used red-fonts to highlight changes seemed to include only some of the passages that were altered, particularly since consideration of models that included 2 types of Ca binding sites were, in some cases, not highlighted by red.The manuscript and new changes still suffer from abundant English usage errors.

We hope that, this time, language errors have been fixed.

The new Figure 1 nicely addresses the issue of what is actually being measured and whether the Q_C_measurements might include contamination from later voltage sensor movements. Furthermore, the figure address the topic of the Ca^2+^ dependent of Q_C_kinetics. There is one issue that persists in this data. The time constant in 1A (0 Ca^2+^) is 52 μs, while in 1B (100 μM) it is 42 μs, both at +160 mV. Examination of the averaged data in panel C show that at +160 mV, μI_G-ON_ is a bit larger than 0.06 ms for 0 Ca^2+^, generally consistent with the trace in 1A, while for 100 μM it appears to be a little above 0.025 ms. The values for the displayed example at 100 μM seems to be well outside the error bars, than one might expect. Perhaps something needs to be checked here.

This issue was also raised by reviewer #1. We agree with reviewer #3 that we made a bad choice of gating current records in Figure 1 A-B. We reviewed our results and the gating current recordings presented in the new Figure 1A-B are representative of the average gating current records.

[Editors' note: further revisions were requested prior to acceptance, as described below

Reviewer #1:The revised manuscript is improved and adequately addresses most of my concerns. However, the authors expressed some confusion over my questions regarding the apparent discrepancy in OFF gating current kinetics, which I will attempt to clarify."We are somewhat confused about the reviewer 1 comment (3). As stated in Results the 𝜏𝐼𝐺−𝑂𝑁(𝑉) curves were fitted to a two-state model (𝜏(𝑉)=1/(𝛼(𝑉)+𝛽(𝑉)) where the forward (𝛼) and backward (𝛽) rate constants represent the resting-active (R-A) transitions of the voltage sensors determining the equilibrium constant of the VSD activation (𝐽(𝑉)=𝛼(𝑉)𝛽(𝑉)). Our statement that the main effect of Ca^2+^ is to slow down the OFF kinetics is based on the determination of 𝛼(𝑉) and 𝛽(𝑉) using the values of 𝜏𝐼𝐺−𝑂𝑁(𝑉) and J(V). This calculation indicates that (𝑉) increase 10.9-fold when the Ca^2+^ concentration is increased to 100 μM (see legend in Figure 1), in a reasonable agreement with the large allosteric factor E. It is unclear to us why the reviewer is referring to Figure 2—figure supplement 1, a figure that is not directly comparable to Figure 1C."My questions about gating current kinetics (Comments #3 in previous review, #1 in original review) relate in large part to the data in Figure 2—figure supplement 1. I asked whether the difference in gating current kinetics shown for 0 and 10 μm Ca in Figure 2—figure supplement 1 are consistent with a large Q-V shift (as observed in 10 Ca). In particular, I noted that there is a relatively small change from 0 to 10 Ca in the time constant of the fast component of OFF current at -90 mV, which doesn't seem compatible with a large Q-V shift. The fast OFF component at -90 mV should be a relatively direct measure of the backward rate β (i.e. τ~=1/β). If there is a small change in ON kinetics and large Q-V shift, then there should be a large change in β and hence OFF kinetics. The authors make essentially the same prediction based on their fits to the ON kinetics in Figure 1C. But they do not include OFF data to directly confirm the prediction or address my concern."We would like to include the OFF time constant plot requested by the reviewer, but these kind of experiments are in extremely difficult to do at high Ca^2+^,.…"I do not think that measurements of OFF kinetics at multiple voltages are necessary. Measurements at a single voltage like -90 mV should suffice. If OFF kinetics have not or cannot be measured at 100 Ca, then I suggest adding the ON kinetics at 10 Ca to Figure 1C together with OFF data at -90 mV for 0 and 10 Ca. Based on Figure 2—figure supplement 1, such data already exists.

We have complied with the request of reviewer #1. The data the reviewer requested is now included in Figure 2—figure supplement 1 as Figure 2—figure supplement 1D. In Figure 2—figure supplement 1D we have followed the same strategy we employed in Figure 1C plotting in the same graph τ_IG-ON_ and Q_C_/Q_C,MAX_ vs. voltage for 0 and 10 μM Ca^2+^. As you can see, the new figure shows that the difference in gating current kinetics shown for 0 and 10 μM Ca^2+^ in Figure 2—figure supplement 1C are consistent with a large Q-V shift that we obtained in 10 μM Ca^2+^. Please note that the change from 0 to 10 Ca^2+^ in the time constant of the fast component of OFF current at -90 mV is compatible with a large Q-V shift (orange circles).